# Genetics-mediated regulation of intestinal gene expression on microbiome contributes to human disease heritability

Haochuan Wang [1,2,3,5], Chengyu Li[1,2,5], Zhen Hu[4,5], Haonan Feng[1,2,5], Luowei Chen[1,2], Ke Ding[1,2], Jiuhong Nan[1,2], Yuhan Wu[1,2,3], Jinghao Sheng [4✉] & Xushen Xiong [1,2✉]

## Abstract

**The gut microbiome plays fundamental roles in physiological and pathological processes, yet its interaction with host gene expression and contribution to disease remain underexplored. Here, we integrate the genetic regulatory maps of 116 microbial genera with gene expression quantitative trait loci (eQTLs) and DNA methylation QTLs (mQTLs) in three intestinal tissues to dissect host–microbiome interaction. We identify 6088, 5810, and 2398 gene-to-microbiome regulatory loci in the transverse colon, sigmoid colon, and ileum, respectively. Among these, 13.2% of genes show broad regulatory effects on multiple genera, with functional enrichments in developmental, metabolic, and immune-related pathways. Integrative analysis with genome-wide association studies (GWASs) reveals 283 microbiome-dependent disease loci. We observe pleiotropic effects mediated by the gene-to-microbiome regulation at both microbiome and disease layers. Notably, we predict and experimentally validate the suppressive effect of *Allisonella* on depression through regulating bile acid abundance, and the regulation of *Parasutterella* on short-chain fatty acid and its contribution to allergic rhinitis. The gene–microbiome–disease regulatory maps are available at our interactive database (https://xiongxslab.github.io/microbiomeMR/).**

**Keywords** Microbiome; Gut–Organ Axis; Disease Genetics
**Subject Categories** Genetics, Gene Therapy & Genetic Disease; Microbiology, Virology & Host Pathogen Interaction

## Introduction

The human intestinal tract provides a living environment for microorganisms, and the gut microbiome in turn plays fundamental roles in regulating human health and various physiological processes, including immune responses and neurological functions (Tremaroli and Bäckhed, 2012; Lynch and Hsiao, 2019). Previous studies have indicated the associations between host genetics and gut microbial traits (Bonder et al, 2016; Wang et al, 2016; Turpin et al, 2016; Kurilshikov et al, 2021). In addition, studies have revealed the connections between the gut microbiome and other molecular traits in humans, such as fecal levels of short-chain fatty acids (SCFAs) (Sanna et al, 2019) and ABO histo-blood group type (Rühlemann et al, 2021). The gut microbiome is connected to multiple common diseases like inflammatory bowel disease (IBD) (Qin et al, 2012; Imhann et al, 2018), as well as along the gut–organ axes, such as the gut–brain (Foster et al, 2017; Collins et al, 2012) and gut–lung axes (Budden et al, 2017; Song et al, 2024). For the diseases along the gut–brain axis, it has now been shown for neuropsychiatric disorders associated with development such as autism (Morais et al, 2021; Su et al, 2024), mood (Bravo et al, 2011; Kelly et al, 2016), and various neurodegenerative disorders including Parkinson's disease and Alzheimer's disease (Tan et al, 2022; Wang et al, 2019b; Morais et al, 2021). The gut microbiome can also mediate the gut–lung axis and is associated with respiratory diseases such as chronic obstructive pulmonary disease (COPD) and asthma (Budden et al, 2017; Song et al, 2024). Regardless of the essential roles of the microbiome in human developmental and pathological processes, the underlying mechanism of the regulatory paths remains largely untapped.

The host genetics have been revealed to affect the diversity and abundance of the gut microbiota (Sanna et al, 2022; Turpin et al, 2016; Lopera-Maya et al, 2022). For instance, several human genomic loci, such as *ABO*, *LCT*, and *FUT2*, genes are associated with the abundance of several taxa (Sanna et al, 2022). Recent research further indicated that the host genome can shape the gut microbiome through epigenetic regulations, including histone and DNA modifications that subsequently influence the expression of immune-related genes (Woo and Alenghat, 2022; Pepke et al, 2024). In the reverse regulatory direction, host gene expression in the intestinal tissues can lead to changes in the composition of gut microbiota (Pepke et al, 2024). For instance, deletion of *SIRT1* in mouse intestinal tissues influences microbiome composition

[1]The Second Affiliated Hospital & Liangzhu Laboratory, Zhejiang University School of Medicine, 311121 Hangzhou, China. [2]State Key Laboratory of Transvascular Implantation Devices, The Second Affiliated Hospital, Zhejiang University School of Medicine, 311121 Hangzhou, China. [3]Zhejiang University-University of Edinburgh Institute (ZJE), Zhejiang University School of Medicine, Zhejiang University, 314400 Haining, Zhejiang, China. [4]Institute of Environmental Medicine, Zhejiang University School of Public Health, 310058 Hangzhou, Zhejiang Province, China. [5]These authors contributed equally: Haochuan Wang, Chengyu Li, Zhen Hu, Haonan Feng. ✉E-mail: jhsheng@zju.edu.cn; xiongxs@zju.edu.cn

(Wellman et al, 2017), and intestinal epithelial cell (IEC) *HDAC3* expression is associated with the composition of intestinal commensal bacteria (Alenghat et al, 2013). The *VANGL1* gene is highly expressed in the gut tissues and is associated with the abundance of *Sutterellacea* (Bonder et al, 2016). On the other hand, the microbiome composition can vary between the small intestine and colon due to the differences in pH, oxygen level, and antimicrobial peptide abundance, and therefore the tissue specificity of gene-to-microbiome regulation also needs clarification (Hall et al, 2017; Donaldson et al, 2016). We envision that a systematic regulatory landscape between intestinal gene expression and microbiome composition will largely facilitate the investigation of crosstalk between host genetics and gut microbiome.

Several human and mouse studies have identified interactions between host genetics and the microbiome in relation to disease phenotypes (Srinivas et al, 2013; Parks et al, 2013). For example, *NOD2* and *CARD9* risk alleles associated with IBD only become manifest when triggered by the gut microbiome (Jostins et al, 2012; Lamas et al, 2016), indicating that genetic predisposition to diseases can depend on the microbiome. Genes involved in the IBD-associated host-microbial interaction, such as *DUOX2* and *DUOXA2*, are associated with the abundance of microbiota such as *Ruminococcaceae* in the ileum tissue (Lloyd-Price et al, 2019). However, a multi-tissue and multi-layer regulatory map across intestinal gene expression, microbiome composition, and disease risk locus is still lacking, hindering the comprehensive mechanistic understanding of the crosstalk between microbiome and human health and diseases.

Here, we integrate Mendelian randomization (MR) and genetic colocalization approaches to characterize the gene-to-microbiome regulation in three intestinal tissues and to understand the mechanistic roles of the crosstalk between intestinal gene expression and microbiome in human complex diseases. Across 116 microbial genera, we identify a total of 6088, 5810, and 2398 regulatory pairs in the transverse colon, sigmoid colon, and ileum, respectively. Based on the pervasiveness of regulatory effects on the microbiome, we categorize the genes into "broadly-regulating" and "specially-regulating" groups and observe different enrichments of functional pathways. We show that over 50% of gene-to-microbiome regulatory loci are shared by at least two tissues. For the tissue-specific regulatory pairs, only about 25% of tissue specificity arises from the eQTL specificity between tissues, while the vast majority represents the tissue specificity at the gene-to-microbiome level. At the global level, gene-to-microbiome regulatory loci in the transverse and sigmoid colons are strongly enriched for the heritability of diverticulosis, pulmonary embolism, and neurodegenerative diseases such as Parkinson's disease. At the locus level, multi-colocalization analysis unravels 133 gene-to-microbiome-dependent disease loci in the transverse colon, 103 in the sigmoid colon, and 47 in the ileum, respectively. We highlight multiple disease loci that implicate gut–organ axes, including asthma and respiratory traits for the gut–lung axis, and *Allisonella*-mediated depression, bipolar disorder, and insomnia disease loci for the gut–brain axis. Notably, we establish the regulatory axes across *Allisonella*, bile acid (BA), and depression, as well as across *Parasutterella*, SCFA, and allergic rhinitis, respectively, which are further verified using mouse experiments. Collectively, our multi-tissue regulatory maps between intestinal gene expression and microbiome, and the integrative analysis with disease genetics,

provide important mechanistic insights into the regulatory role of microbiome on human health and diseases.

# Results

## Regulatory map between gene regulation and microbiome in intestinal tissues

To elucidate the crosstalk between intestinal gene regulation and gut microbiome composition, we established a statistical genetics framework that integrated bidirectional MR and colocalization analyses (Fig. 1A,B). The eQTLs in transverse colon, sigmoid colon, and ileum, and mQTLs in transverse colon tissues were collected from the GTEx Consortium, and microbiome composition QTLs (mbQTLs) were collected from the MiBioGen (Table EV1) (Kurilshikov et al, 2021; GTEx Consortium, 2020). The sample sizes for the eQTL calling were 368 for the transverse colon, 318 for the sigmoid colon, and 174 for the ileum, and the mbQTLs were identified from the 16S rRNA fecal microbiome data involving 116 genera across 18,340 individuals. For the gene-to-microbiome regulation, we used intestinal eQTLs as instrumental variables (IVs) to delineate the regulation of gene expression (exposure) on the composition of 116 genera (outcome). Conversely, we utilized mbQTLs as IVs for inferring the potential regulation from microbiome to intestinal DNA methylation and gene expression (Fig. 1A). We then carried out MR-Egger and heterogeneity in dependent instruments (HEIDI) tests to ensure the robustness of causal inference and to avoid potential confusion caused by pleiotropic SNPs (Zhu et al, 2016). To further prioritize high-confidence regulatory pairs, we additionally integrated colocalization analysis for each genetic locus (Fig. 1B).

Following the procedure, we identified 6,088 gene-to-microbiome regulatory pairs in the transverse colon, 5810 in the sigmoid colon, and 2398 in the ileum, respectively, based on an adjusted *P* value threshold of 0.05 in the MR analysis (Figs. 1B,C and EV1A; Dataset EV1). To assess the reliability of the identified regulatory pairs, we leveraged the transverse colon eQTLs from the CEDAR database as an independent cohort for external validation (Momozawa et al, 2018; Data ref: Momozawa et al, 2018). Of the 6088 regulatory pairs we identified in transverse, 767 gene–microbiome regulatory pairs were tested due to the availability of significant eQTLs as IVs in the CEDAR data. We found that 295 (38.5%) regulatory pairs can be independently validated in the replication dataset, with a strong consistency in effect size (Fig. EV1B, $R^2 = 0.68$, *P* value $= 2.9 \times 10^{-77}$), regardless of the different statistical power and different cohorts between GTEx and CEDAR eQTLs.

Each microbial genus on average possesses 52, 50, and 21 regulatory pairs in the transverse colon, sigmoid colon, and ileum, respectively (Fig. EV1C). To understand the magnitude of gene-to-microbiome regulation, we investigated the effect sizes of the identified regulatory pairs in the three tissues. In the transverse colon, *Lactococcus*, *Gordonibacter*, and *Methanobrevibacter* were identified as the top genera receiving the strongest overall regulation from host gene expression (Fig. 1D). In total, 14 (70%) of the top 20 genera from the three tissues were overlapped, indicating the consistent regulatory effects of intestinal gene expression on microbiota across three tissues (Figs. 1D and EV1D).

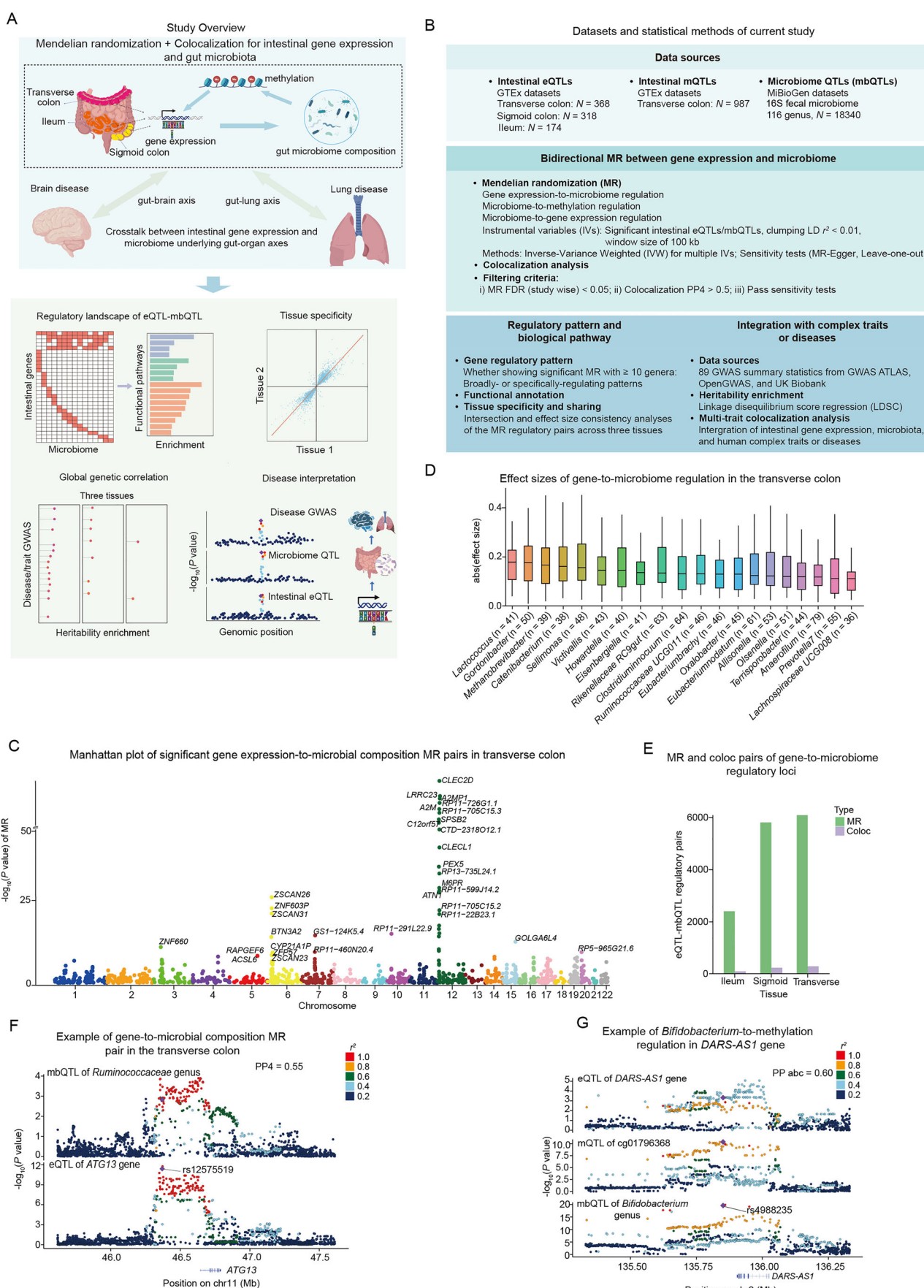

**A** Study Overview
Mendelian randomization + Colocalization for intestinal gene expression and gut microbiota

**B** Datasets and statistical methods of current study

**C** Manhattan plot of significant gene expression-to-microbial composition MR pairs in transverse colon

**D** Effect sizes of gene-to-microbiome regulation in the transverse colon

**E** MR and coloc pairs of gene-to-microbiome regulatory loci

**F** Example of gene-to-microbial composition MR pair in the transverse colon

**G** Example of *Bifidobacterium*-to-methylation regulation in *DARS-AS1* gene

◀ **Figure 1. Genetics-based regulatory landscape between intestinal gene expression and microbiome composition.**

(A) Sketch plot showing the overview of this study. The top panel illustrates the identification of crosstalk between intestinal gene expression, DNA methylation, and gut microbiome composition. In the forward direction, the regulation from intestinal gene expression to microbiome was tested. In the reversed direction, the regulation of microbiome composition to DNA methylation and to gene expression was tested. The bottom panel shows the multiple downstream interpretations based on the regulatory landscape. (B) Detailed information about the QTL datasets used for identifying the gene-to-microbiome, microbiome-to-methylation, and microbiome-to-gene regulation, and the step-by-step procedure of the statistical methods. (C) Manhattan plot illustrating the genome-wide significant gene-to-microbiome regulatory pairs in the transverse colon. Each dot represents a gene locus with regulatory effects on the microbiome, with the x axis indicating the genomic position and the y axis representing the log-changed P value. The top regulatory loci are labeled. (D) The effect sizes of the gene-to-microbiome regulation of the top 20 genera in the transverse colon. The y axis indicates the absolute effect size, and the x axis indicates the top 20 genera with the strongest overall regulatory magnitude. Each box's central line indicates the median, and the bounds of the box represent the 25th and 75th percentiles (interquartile range, IQR). The whiskers extend to the most extreme data points that are within 1.5 × IQR from the box. The value of n shown in each x axis label represents the number of gene-to-microbiome regulatory pairs used for that genus. (E) Summary of significant MR-based gene–microbiome pairs, colocalization pairs, and the overlapped pairs in the ileum, sigmoid colon, and transverse colon. (F) MR and colocalization evidence for the regulatory effect of *ATG13* locus (rs12575519) on the *Ruminococcaceae* genus in the transverse colon. The data points denote genetic variants with the lead SNP of the colocalization highlighted, and are colored by the linkage disequilibrium ($R^2$) with the lead SNP. The posterior probability for colocalization is 0.55. (G) LocusCompare visualization of the multi-trait colocalization across *DARS-AS1* gene expression, methylation of cg01796368, and *Bifidobacterium* genus in the transverse colon. The y axes show log-changed P value of association tests in eQTL, mQTLs, and mbQTLs, respectively. The lead variant is indicated, and the correlation of the LD effect relative to the lead variant is represented by the color scale. Source data are available online for this figure.

Intriguingly, we observed a pronounced regulatory cluster in chromosome 12 (Figs. 1C and EV1A), and found that the underlying genes were strongly enriched in pathways relevant to intestinal and microbial functions, such as regulation of calcium-mediated signaling (Loh et al, 2024), sensory perception of taste (Depoortere, 2014), and serine family amino acid catabolic process (Mardinoglu et al, 2015; Li et al, 2024a) (Fig. EV1E). Intriguingly, while previous studies have reported that chromosome 12 harbors multiple genetic risk loci linked to intestinal diseases, including Crohn's disease (CD) and ulcerative colitis (UC), our result provides a potential microbiome-dependent regulatory circuit underlying these diseases (Herrick and Tansey, 2021; Lesage et al, 2000; Lin et al, 2020).

For the gene-to-microbiome regulatory pairs identified by MR, we further performed colocalization analysis (Fig. 1B, "Methods"). We found 285, 234, and 91 colocalized pairs (PP4 > 0.5) in the transverse colon, sigmoid colon, and ileum, respectively (Fig. 1E; Dataset EV2), with a total of 22 pairs showing strong posterior colocalization probabilities (PP4 > 0.9, defined as tier 1). Of the 285 colocalization loci identified in the transverse colon, 37 with significant eQTLs in CEDAR cohort were further examined independently, and 19 (51%) colocalized pairs were reproducible with this external dataset. We show an example locus supported by both MR (P adjust = $3.18 \times 10^{-5}$) and colocalization (PP4 = 0.55) evidence, where the gene expression level of *ATG13* in the transverse colon is predicted to regulate the abundance of *Ruminococcaceae* genus (Fig. 1F). *ATG13* is an essential gene for the initiation of autophagy (Hao et al, 2022; Wang et al, 2024), and studies have reported a decreased abundance of the *Ruminococcaceae* genus in IBD mice with enhanced autophagy signatures, collectively suggesting an association between autophagy activity and *Ruminococcaceae* abundance (Larabi et al, 2020; Wang et al, 2019a). Our finding further provides a potential mechanistic link connecting the intestinal *ATG13* expression, *Ruminococcaceae* genus, and the autophagy process (Fig. 1F).

We next interrogate the opposite direction by searching for the regulatory loci from gut microbiome composition to gene expression. We detected 46 significant microbiome-to-gene regulatory pairs in the transverse colon, 45 in the sigmoid colon, and 34 pairs in the ileum, respectively, at an adjusted P value threshold of 0.05 in the MR analysis (Fig. EV1F; Dataset EV3). Among these

regulatory pairs, 3, 6, and 3 loci also displayed genetic colocalization effects in the three tissues, respectively (Fig. EV1F; Table EV2). Since microbiome composition may affect host gene expression with epigenetic regulation as an intermediate mechanism (Pepke et al, 2024; Woo and Alenghat, 2022), we further carried out MR and colocalization analyses between microbiome and DNA methylation ("Methods"). We identified 267 microbiome-to-methylation regulatory pairs using the MR approach, with 35 additionally showing colocalization effects in the transverse colon (Fig. EV1G; Dataset EV4; Table EV3). To establish the potential regulatory path across microbiome, DNA methylation, and gene expression, we performed multiple-trait colocalization and identified 28 loci for these three layers in the transverse colon (Table EV4). For instance, we noticed a strong multi-trait colocalization signature across the *Bifidobacterium* genus, DNA methylation site (cg01796368), and *DARS-AS1* gene in the transverse colon (Fig. 1G). Previous studies have demonstrated that the *Bifidobacterium* genus can activate autophagy (Engevik et al, 2019) and that *DARS-AS1* gene can also regulate cytoprotective autophagy (Wang et al, 2024). Our results further reveal a potential mechanistic link, which suggests that the *Bifidobacterium* genus may influence *DARS-AS1* gene expression through DNA methylation.

The substantially smaller number of loci identified for the regulatory direction from microbiome composition to intestinal gene expression, compared to the opposite direction, may be in part attributable to the relatively lower heritability of microbial composition (Morris and Bohannan, 2024). To ensure sufficient statistical power and reliability, we primarily focused on the gene-to-microbiome regulation for downstream functional interpretation.

## Gene and pathway specificity of gene-to-microbiome regulation

To understand the specificity and sharing of microbiome genera regulated by the expression of genes, we first summarized the frequency of genes regulating different numbers of microbiota (Figs. 2A and EV2A). While the vast majority of genes can only regulate a very limited number of genera, there are several genes showing pleiotropic regulatory effects on a broad range of genera.

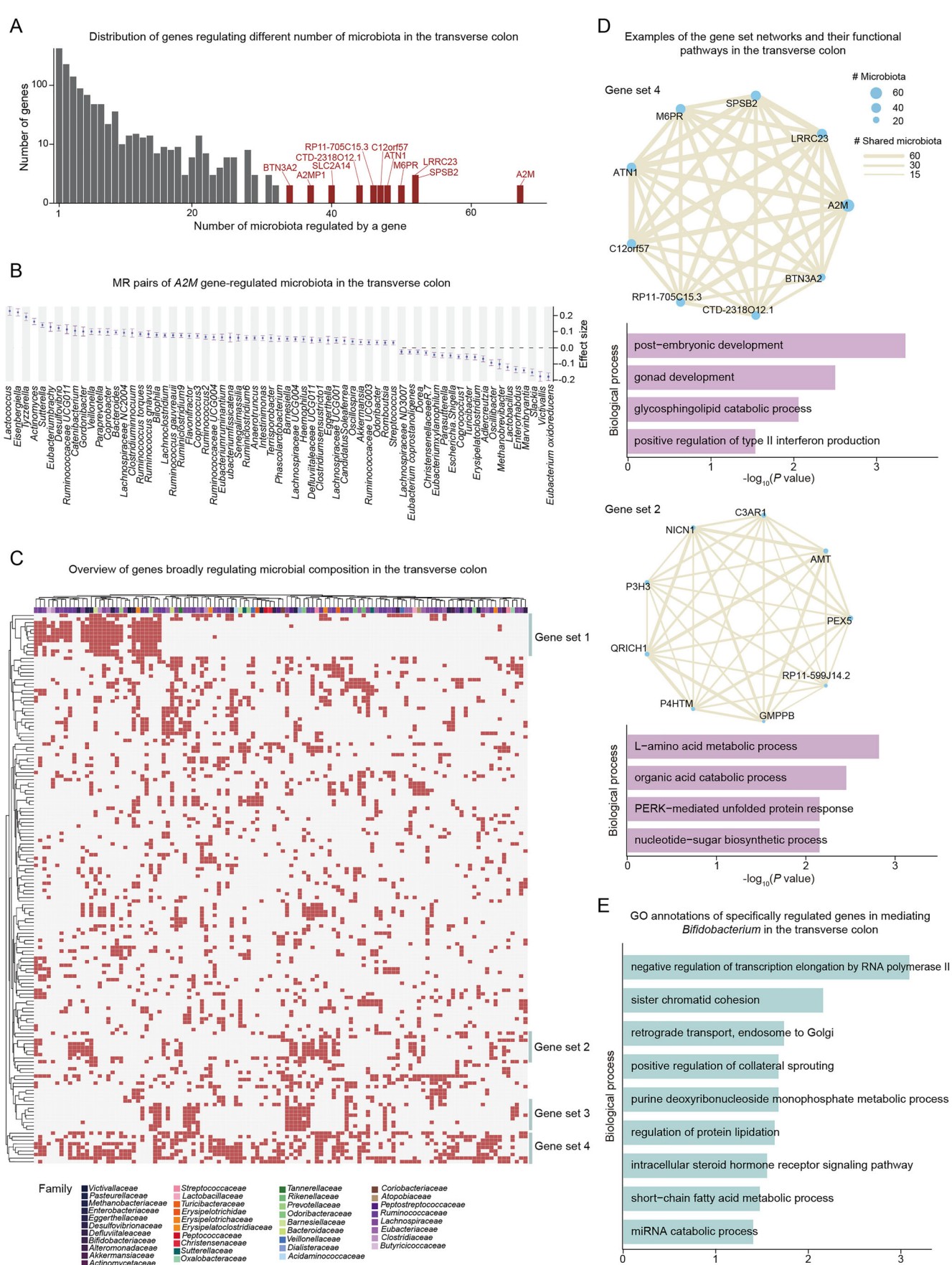

A — Distribution of genes regulating different number of microbiota in the transverse colon

B — MR pairs of *A2M* gene-regulated microbiota in the transverse colon

C — Overview of genes broadly regulating microbial composition in the transverse colon

D — Examples of the gene set networks and their functional pathways in the transverse colon

E — GO annotations of specifically regulated genes in mediating *Bifidobacterium* in the transverse colon

◀ **Figure 2. Pathway annotation of microbiome-regulating genes.**

(A) Bar plot showing the distribution of the number of genes based on the number of microbiota regulated by each gene in the transverse colon. Genes regulating more than 35 microbiota genera were highlighted in red with gene names labeled. (B) An example showing the effect sizes of *A2M* gene expression in the transverse colon on the gut microbiome composition across 67 microbial genera. (C) Heatmap showing the genes that broadly regulate multiple microbiota (more than 10 genera) in the transverse colon, with four gene sets identified based on hierarchical clustering. (D) Networks of genes formed by gene set 4 (top) and gene set 2 (bottom) were identified in the transverse colon, where each node represents the number of microbiota regulated by the genes, and each edge indicates the shared microbiota between genes. The bar plots underneath show the pathway enrichments for these two gene sets. Functional enrichments were calculated by using Fisher's exact test. (E) Functional enrichments of the genes that specifically regulate *Bifidobacterium* genus. Functional enrichment was calculated by using Fisher's exact test. Source data are available online for this figure.

For instance, *A2M*, *SPSB2*, and *LRRC23* genes in the transverse colon are predicted to modulate the composition of over 40 different genera in the gut microbiome (Fig. 2A). As an example, we show the MR regulatory map for *A2M* gene, which encodes a macroglobulin involved in the inflammatory cascades (Ramlall et al, 2020), and was predicted to affect 67 genera of the microbiome in the transverse colon (Fig. 2B). *A2M* gene expression was predicted to show the strongest positive regulatory effects on *Lactococcus*, *Eisenbergiella*, and *Tyzzerella*, and the strongest negative effects on *Enterorhabdus*, *Slackia*, and *Victivallis*, respectively (Fig. 2B). *A2M* gene has been previously suggested to be involved in multiple intestinal diseases, including colorectal cancer (CRC), IBD, and irritable bowel syndrome (IBS) (Goo et al, 2012; Lagrange et al, 2022; Desai et al, 2024), and our result further established a potential mechanistic link mediated by microbiome regulation.

We next sought to categorize genes based on their pervasiveness in regulating the microbiome composition and defined the genes regulating more than 10 different genera as a "broadly-regulating" group ("Methods"). The genes involved in the "specifically-regulating" groups showed significantly higher regulatory effect size compared to those in the "broadly-regulating" groups in all three tissues (Fig. EV2B). We then further explored the sharing and specificity of regulation between genes and microbiome via a hierarchical clustering analysis, and noted four gene sets in the transverse colon that show clustering patterns, where genes in set 4 can broadly regulate nearly all families of the microbiome, and genes in set 1 to 3 can each regulate a specific cluster of families (Fig. 2C). In addition to the transverse colon, we also observed four gene sets in the sigmoid colon and three gene sets in the ileum (Fig. EV2C,D). We compared the sharing of genes across the gene sets identified in the three tissues by the Jaccard index, and observed strong gene similarities (0.24 to 0.77) between the gene sets across different tissues (Fig. EV2E).

In the transverse colon, we show that genes in set 4 form a highly interconnected network with the sharing of regulating genera, and are strongly enriched for fundamental biological processes, including post-embryonic and gonad development, as well as interferon production, in line with the known roles of the microbiome in various immunology pathways (Zheng et al, 2020; Maloy and Powrie, 2011; Ivanov and Honda, 2012) (Fig. 2D). Genes in set 2 are enriched for amino acid metabolic and organic acid catabolic pathways, which is consistent with previous functional studies of gut microbiome in reshaping the host amino acid landscape (Joly and De Vadder, 2024; Li et al, 2024a) (Fig. 2D). In addition to the genes with broad effects on multiple genera, we also looked into the functional enrichments for the genes that exert

a more specific regulation on a smaller number of genera (Dataset EV5). As examples, we show the functional pathways for genes that specifically regulate *Bifidobacterium* and *Oscillibacter* (Figs. 2E and EV2F), both of which are associated with the production of fatty acid (Tsukuda et al, 2021; Martínez-Cuesta et al, 2021). Consistently, we found significant enrichments of SCFA metabolic processes for the genes regulating *Bifidobacterium* and *Oscillibacter*, along with other metabolic pathways such as monophosphate and glycine metabolic processes (Figs. 2E and EV2F).

## Tissue specificity and sharing of the microbiome regulation

To investigate the tissue specificity of the microbiome regulation by gene expression, we performed an intersection of the gene-to-microbiome regulatory pairs across the three intestinal tissues (Fig. 3A,B). We found that 3611 pairs are shared by at least two tissues, and that 18.8% in the ileum, 44.4% in the sigmoid colon, and 44.9% in the transverse colon show tissue-specific regulatory effects on microbiome composition (Fig. 3B). We compared the effect sizes in different "tissue-sharing" and "tissue-specific" groups, and observed that genes involved in the transverse-colon-specific group exhibited the overall strongest regulatory effects, while the genes in the group with regulatory pairs shared by three tissues showed the overall weakest effects Fig. EV3A).

Since the MR analysis was based on the genetic regulations of both gene expression and microbiome composition, we next investigated whether the tissue-specific MR pairs were due to the tissue specificity of eQTL or the specificity of regulation. We found that only 15.7–26.4% of tissue specificity arose from eQTL specificity, whereas the eQTLs in the remaining 73.6–84.3% tissue-specific pairs were shared across tissues, indicating real specificity of regulation from gene expression to the microbiome (Fig. 3C). In addition to the intersection analysis, we further assessed the regulatory directionality and effect size for these tissue-specific regulatory pairs and found high collinearities ($R^2 = 0.90$–$0.92$) across the three tissues. Regarding the effect directionality, we found that 99.3–100% of regulatory pairs were consistent between tissues, indicating the robustness of gene expression's effect on microbiome composition (Figs. 3D and EV3B). We further examined the effect size consistency across tissues for the tissue-specific regulatory pairs. While these pairs were statistically significant in only one tissue, their effect sizes and directionalities remained consistent with other tissues ($R^2 = 0.78$, Fig. EV3C). Notably, the regression coefficient was 0.81, suggesting that the effect sizes in the non-significant tissues were relatively

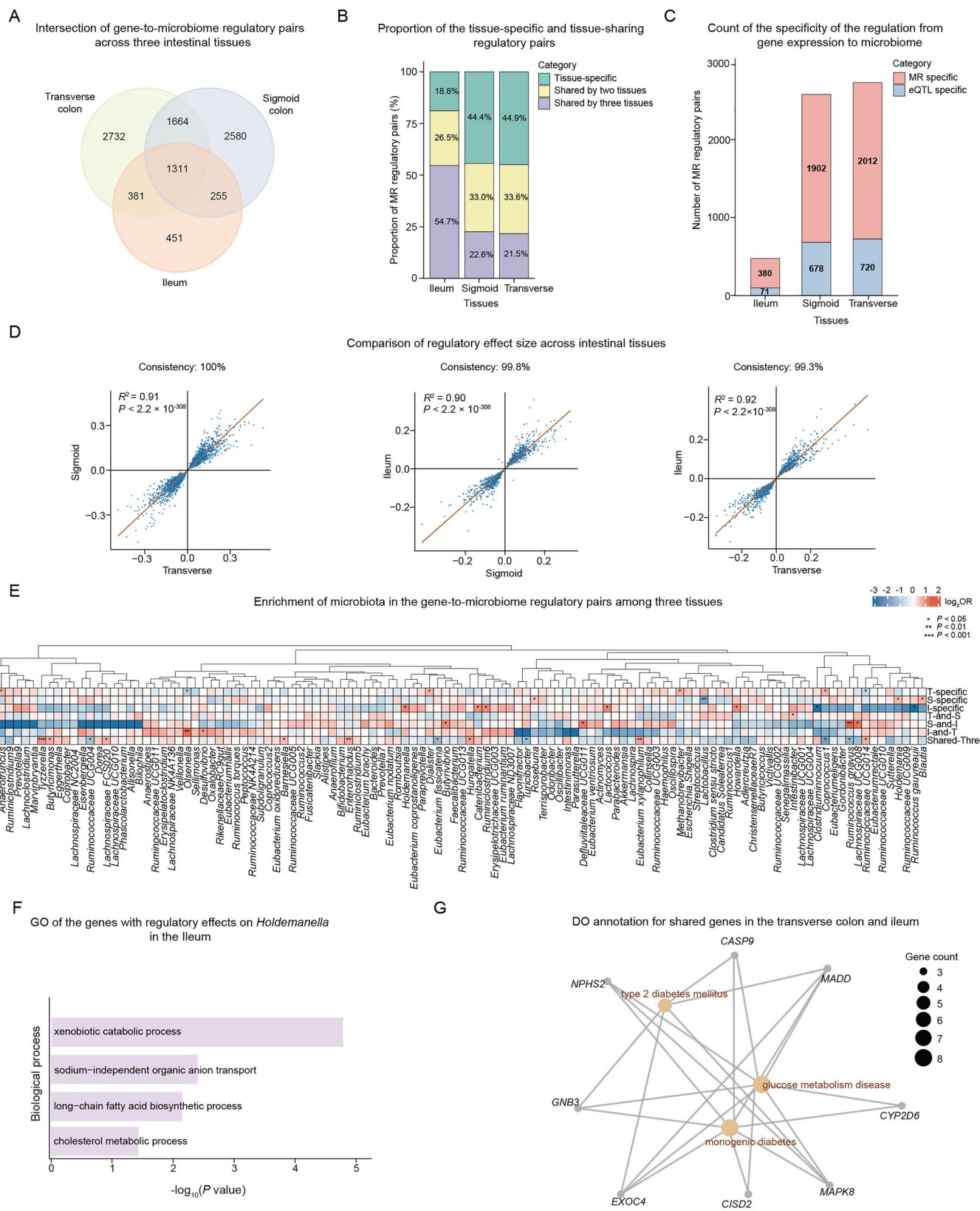

**A** Intersection of gene-to-microbiome regulatory pairs across three intestinal tissues

**B** Proportion of the tissue-specific and tissue-sharing regulatory pairs

**C** Count of the specificity of the regulation from gene expression to microbiome

**D** Comparison of regulatory effect size across intestinal tissues

**E** Enrichment of microbiota in the gene-to-microbiome regulatory pairs among three tissues

**F** GO of the genes with regulatory effects on *Holdemanella* in the Ileum

**G** DO annotation for shared genes in the transverse colon and ileum

**Figure 3. Tissue specificity of the gene-to-microbiome regulation.**

(A) Venn diagram indicating the intersection of the gene-to-microbiome regulatory pairs across the three intestinal tissues. The regulatory pairs can be categorized into seven groups based on the intersection. (B) Proportion of the tissue-specific regulatory pairs, and the pairs shared by two and three tissues in each tissue. (C) Number of tissue-specific regulatory pairs at the eQTL-specific level and the real gene-to-microbiome level in the three tissues. (D) Pairwise comparisons of the effect sizes of the gene-to-microbiome regulatory pairs across three tissues. Pearson correlation coefficients and the corresponding $P$ values were calculated. The directionality consistency for each pairwise comparison is shown at the top of each panel. (E) Enrichment of microbiota in the gene-to-microbiome regulatory loci across the seven tissue-shared and tissue-specific categories. The red blocks represent enrichment, and the blue blocks represent depletion. The $P$ values were calculated using a two-sided Fisher's exact test, and the significant enrichments were indicated using asterisks. I, T, and S represent the ileum, transverse colon, and sigmoid colon, respectively. (F) Functional enrichments of the regulatory loci in the ileum in influencing the *Holdemanella* genus. Functional enrichment was calculated by using Fisher's exact test. (G) A network showing the disease ontology (DO) analysis of the regulatory loci shared by the transverse colon and ileum. The dot size represents the number of genes involved in the corresponding disease. Source data are available online for this figure.

weaker compared to the tissue where significance was detected. These results indicated that these tissue-specific regulatory pairs maintain a certain degree of biological consistency in other tissues, but with a lower effect magnitude.

We then calculated the enrichments of MR pairs for different genera in the seven groups with different tissue specificity and sharing (Fig. 3E). We noticed that genera such as *Methanobrevibacter* and *Dialister* were enriched for the transverse-colon-specific MR regulatory pairs, *Holdemania* and *Blautia* were enriched for sigmoid-colon-specific pairs, and *Holdemanella* and *Howardella* were enriched for the ileum-specific MR group (Fig. 3E). We also noted pronounced enrichments for several genera that had MR pairs shared by pairs of the tissues, including *Intestinibacter* regulation shared by transverse and sigmoid colons, *Ruminococcus gnavus*, *Lachnospiraceae UCG008* shared by sigmoid colon and ileum, and *Olsenella* and *Desulfovibrio* shared by ileum and transverse colon (Fig. 3E). The intestinal mucus layers are built around highly glycosylated gel-forming mucin (MUC2) secreted by goblet cells, and the glycan present in mucins are of diverse and complex structure consisting of four core mucin-type O-glycans (Owen et al, 2017). Notably, *Ruminococcus gnavus* was strongly enriched for the regulatory pairs shared by sigmoid colon and ileum, consistent with the previous report that the sigmoid colon and ileum contain MUC2 O-glycan and that *Ruminococcus gnavus* degrades MUC2 to utilize mucin glycans which is served as metabolic substrates (Crost et al, 2016; Owen et al, 2017; Tailford et al, 2015; Johansson et al, 2011). While the regulation on several genera such as *Tyzzerella* and *Hungatella* were enriched across all three tissues, genera including *Eubacterium fissicatena* and *Turicibacter* tend to be depleted for the tissue-sharing regulation (Fig. 3E).

We then carried out functional analyses for the MR regulatory pairs that show enrichments in the microbiome-tissue specificity analysis (Dataset EV6). For the genes involved in the tissue-sharing microbiome regulation, pathways related to cell cycle, development, and metabolism show the highest frequency in the GO analysis, while pathways pertaining to metabolism, transport, and localization are among the top terms in the tissue-specific regulatory groups (Fig. EV3D). For the genes that regulate the *Holdemanella* genus in the ileum, we found significant enrichments in the long-chain fatty acid biosynthetic process (Fig. 3F), in line with the previous report (Pujo et al, 2021). To further provide disease implications for the gene-to-microbiome regulation, we performed Disease Ontology (DO) analyses on the involved genes (Figs. 3G and EV3E; Table EV5). Intriguingly, for the genes showing regulatory effects on the microbiome shared by the ileum and

transverse colon, we found strong enrichments in multiple metabolic-related diseases, such as diabetes mellitus and glucose metabolism disease (Fig. 3G).

## Gene-to-microbiome regulation contributes to disease heritability

Since the gut microbiome has been revealed to play important roles in various human diseases, we next sought to establish the genetics-dependent regulatory circuits from intestinal gene expression to microbiome composition, and eventually to human complex diseases and phenotypes. We collected 89 GWAS datasets from GWAS Atlas, Open GWAS, and UK Biobank, focusing on the diseases or traits that could be relevant to the microbiome ("Methods", Dataset EV7). We first evaluated the enrichments of the gene-to-microbiome regulation in the various GWAS traits by heritability partitioning in three intestinal tissues. In the sigmoid colon, the regulatory pairs were enriched in 14 genetic diseases and traits ($P$ value < 0.05, Fig. 4A), including multiple respiratory traits and diseases, encompassing forced vital capacity (FVC), forced expiratory volume in 1 s (FEV1), and lung cancer, gastrointestinal diseases such as diverticulosis, and neurodegenerative diseases such as Parkinson's disease. In the transverse colon, we observed disease enrichments that were similar to those in the sigmoid colon, in line with the biological similarity between the two tissues. In the ileum, we observed enrichments in a limited number of GWAS traits, potentially due to the relatively fewer MR pairs identified in this tissue (Fig. 4A).

In addition to the enrichment analysis, we further identified the GWAS loci dependent on the gene-to-microbiome regulation for each tissue using multi-trait colocalization (moloc) (Giambartolomei et al, 2018) and two-step MR analyses ("Methods"). We characterized 133 multi-trait colocalization disease loci in the transverse colon, 103 loci in the sigmoid colon, and 47 loci in the ileum, respectively, with 76 loci further supported by two-step MR analyses (Fig. EV4A; Dataset EV8). We asked if specific genera play dominant roles in mediating the regulation between gene expression and GWAS loci, by counting the frequency of their disease-associated loci (Figs. 4B and EV4B). In the transverse colon, *Allisonella*, *Blautia*, and *Dorea*, which belong to different families but were all suggested to be associated with SCFA production (Ecklu-Mensah et al, 2023; De Bruyn et al, 2024; Baxter et al, 2019), are among the top genera that possessed the largest number of significant regulatory pairs (Fig. 4B). To confirm the causal roles of these genera in SCFA production, we carried out bacterial colonization experiments in mice and examined SCFA levels. We

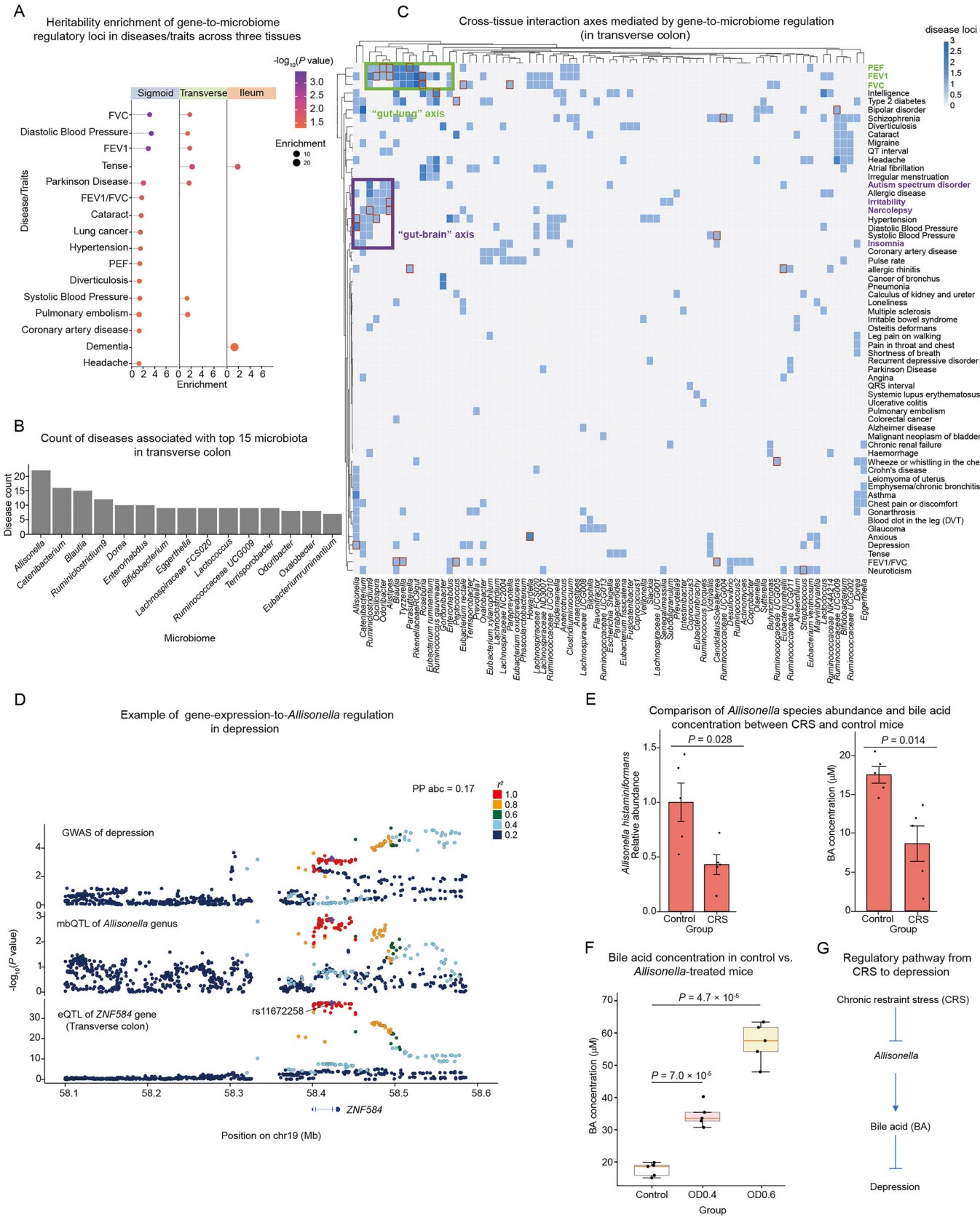

◄ **Figure 4. Gene-to-microbiome regulation contributes to disease heritability.**

(A) Heritability enrichment of the gene-to-microbiome regulatory loci in the GWAS of complex diseases or traits by linkage disequilibrium score regression (LDSC). The enrichment *P* value is shown in a color scale; the enrichment folds and *P* values are shown for those traits with *P* value < 0.05. Enrichment *P* value was calculated by S-LDSC via z-score calculation. (B) The number of genetic diseases and traits (*y* axis) associated with different microbiota (*x* axis) identified through multi-trait colocalization analysis in the transverse colon. (C) Heatmap showing multi-trait colocalization between complex diseases (row) and the genera (column) involved in gene-to-microbiome regulation pairs in the transverse colon, with the color scale representing the number of loci. The red boxes represent the loci that were further supported by two-step MR analyses. FVC represents forced vital capacity, FEV1 represents forced expiratory volume in 1 s, and PEF represents peak expiratory flow. (D) LocusCompare visualization of the multi-trait localization across depression GWAS, *Allisonella* genus mbQTL, and *ZNF584* eQTL in the transverse colon. The *y* axis shows the log-changed *P* value of association tests in GWAS, mbQTLs, and eQTLs. The lead variant is indicated, and the correlation of the LD effect relative to the lead variant is represented by the color scale. PP abc is indicated at the top right, which represents the posterior probability of colocalization across GWAS, mbQTLs, and eQTLs. (E) *Allisonella* species abundance (left) and bile acid (BA) level (right) in the chronic restraint stress (CRS)-treated and control mice. *P* value was calculated by using unpaired *t* test. Each group contains five replicates. Error bars indicate standard error of the mean. (F) Comparisons of the BA concentration between control mice and mice treated with different concentrations of *Allisonella*. The box's central line indicates the median. The bounds of the box represent the 25th and 75th percentiles (interquartile range, IQR). The whiskers extend to the most extreme data points that are within 1.5 × IQR from the box. *P* value of each comparison was calculated using unpaired *t* test. Each group contains five replicates. (G) Schematic illustrating the regulatory axis across CRS treatment, *Allisonella* abundance, BA level, and depression. Source data are available online for this figure.

selected one representative strain for each genus, and found that all three colonized mouse groups exhibited significantly upregulated SCFA levels (Fig. EV4C).

We then investigated the potential relationship between GWAS traits and different genera in the microbiome, via a clustering analysis based on the number of regulatory pairs (Fig. 4C, "Methods"). Notably, respiratory-related traits, including FVC, FEV1, and peak expiratory flow (PEF) formed a regulatory cluster with various genera, suggesting a "gut–lung" axis given the known functions of the gut microbiome in regulating the immune system and subsequently the susceptibility to respiratory function and diseases. Moreover, we noted a GWAS cluster of various brain-related diseases and phenotypes, including autism, irritability, narcolepsy and insomnia, that was dependent on the regulation of *Allisonella, Catenibacterium, Oscillospira, Odoribacter*, and *Alistipes*, which have been revealed to participate in SCFA production and immune functions (Huang et al, 2023; Yang et al, 2021; Chanda and De, 2024; Older et al, 2024; Xing et al, 2021; Wu et al, 2025), providing mechanistic insights into the "gut–brain" axis mediated by the gene-to-microbiome regulation (Fig. 4C). Likewise, we also observed such cross-tissue interactions, including gut–brain and gut–lung axes, mediated through the gene-to-microbiome regulation in sigmoid colon and ileum (Fig. EV4D,E).

We observed a multi-trait colocalization signature across *ZNF584* gene, *Allisonella* genus, and depression GWAS (Fig. 4D), which was further supported by our two-step MR analysis. Given the previously established role of BA in depression (Klein and Kheirbek, 2024; Li et al, 2024b) and our predicted relationship between *Allisonella* and depression, we next explored the potential causal chain from *Allisonella* composition to BA production, and eventually to depression. We first treated mice with chronic restraint stress (CRS), a commonly used strategy for constructing depression mouse models, and observed significant reductions in both *Allisonella* abundance ($P = 0.028$) and BA level ($P$ value = 0.014) (Fig. 4E), consistent with the known suppressive effect of BA on depression. We further explored the relationship between *Allisonella* abundance and BA level by mouse oral gavage of *Allisonella* with different concentrations ($OD_{600} = 0.4$ and 0.6), and found that *Allisonella*-treated mice exhibited dosage-sensitive increase of BA levels compared to the control group (Fig. 4F). Collectively, building on the established role of BA in suppressing

depression, our analyses and experiments further filled a regulatory gap where CRS treatment contributes to depression via *Allisonella*-mediated regulation of BA production (Fig. 4G).

Multi-trait colocalization and two-step MR analyses also revealed the relationship across *ABHD12* gene, *Parasutterella*, and allergic rhinitis (Fig. EV4F). Intriguingly, previous studies have indicated the association between *Parasutterella* abundance and allergic rhinitis (Yang et al, 2022), as well as the regulatory role of SCFAs in allergic rhinitis (Zhou et al, 2021; Liu et al, 2023). However, the exact relationships across *Parasutterella*, SCFA, and allergic rhinitis remained unclear given the regulatory gap between *Parasutterella* and SCFA. To address this, we performed bacteria-colonization experiments in mice using two different concentrations of *Parasutterella* treatments. We observed significantly increased levels of acetate, a major type of SCFA, in the *Parasutterella*-treated mice, with the higher treatment concentration inducing a stronger SCFA elevation (Fig. EV4G, $P$ value = $1 \times 10^{-3}$ for OD0.4 and $4.9 \times 10^{-5}$ for OD0.6). Therefore, our results established the regulation of *Parasutterella* on SCFA, providing a mechanistic link to their regulatory roles in allergic rhinitis (Fig. EV4H).

We also noted more complex regulatory modes where multiple genera were involved in the same gene-to-disease regulation, suggesting a pleiotropic effect at the microbiome level. For instance, in the "gut–brain" axis, we found that *Bifidobacterium* (PP abc = 0.75), *Ruminococcaceae UCG002* (PP abc = 0.37), and *Ruminococcaceae UCG009* (PP abc = 0.48) all show strong colocalizations with schizophrenia GWAS and *ATG13* (Fig. EV5A), which encodes an autophagy-related protein involved in the phagosome formation (Bhattacharya et al, 2024). In addition, the *CDK2AP1*-to-*Allisonella* regulatory pair can explain the heritability of multiple diseases, including asthma, allergic rhinitis, and FEV based on the multiple colocalization analysis (Fig. EV5B), indicating pleiotropic effects of gene-to-microbiome regulation in mediating multiple diseases. These representative examples collectively demonstrated different modes of regulation between gene expression and microbiome composition in mediating lung and brain diseases.

To facilitate the exploration of the relationship across intestinal gene expression, microbiome composition, and disease heritability, we compiled our results into an interactive web server that is publicly available (https://xiongxslab.github.io/microbiomeMR/).

# Discussion

Our study presents a multi-tissue interaction map between intestinal gene expression and gut microbiome composition, establishing the regulatory path from gene expression to 116 microbial genera. We explored the functional enrichments and tissue specificity of the gene-to-microbiome regulatory pairs. By further integrating with GWAS datasets, we constructed the gene–microbiome-disease regulatory loci, shedding light on the microbiome-dependent mechanism for hundreds of disease loci indicative of the gut–brain and gut–lung axes.

While the microbiome has been revealed to be influenced by the host genetics, only a few loci such as *LCT*, *ABO*, and *FUT2* have been consistently confirmed to modulate the microbiome composition (Sanna et al, 2022). The eQTLs of intestinal tissues and microbiome QTLs provide ideal genetic instruments to delineate the interaction between gene expression and the microbiome. Leveraging the MR approach, we identified 6088 gene-to-microbiome regulatory pairs in the transverse colon, 5810 in the sigmoid colon, and 2398 in the ileum, respectively, substantially expanding the repertoire of regulatory pairs. The relatively smaller number of regulatory pairs identified in the ileum was likely due to the smaller eQTL sample size ($N = 174$) compared to the transverse colon ($N = 368$) and sigmoid colon ($N = 318$). In addition to MR, we further integrated the genetic colocalization method to provide additional support for the regulation. We characterized 610 colocalization loci between intestinal gene expression and microbiome.

We defined genes with broad regulatory effects on multiple genera and consistently identified three to four gene sets with different functional enrichments for each tissue. The genes from the "broadly-regulating" groups are enriched for various fundamental pathways, including developmental processes, immune-related, and metabolic-related pathways, consistent with the reported functions of the microbiome in governing gut-related phenotypes. In addition to the genes with broad effects, we also investigated the pathway enrichments for the genes with more specific effects on a limited number of genera. We show the functions of the genes that specifically influence *Bifidobacterium* and *Oscillibacter* as examples and found strong enrichments in metabolic processes relevant to SCFAs. Collectively, these pathway enrichment results together reveal both known and previously unappreciated functions of the microbiome.

The variation in microbiome composition between intestinal tissues has been reported (Hall et al, 2017; Donaldson et al, 2016), and therefore we also systematically evaluated the tissue specificity of the gene-to-microbiome regulation across the ileum, transverse colon, and sigmoid colon. We demonstrated that about 19–45% of the regulatory pairs are specific to one of the three tissues. Further examination of the source of specificity, we found that the vast majority of tissue-specific loci stem from the regulation of gene expression on the microbiome rather than from the specificity of eQTLs. Of note, given that the tissue-specific regulatory pairs maintained an overall consistent but reduced effect size in the non-significant tissues, the observed tissue specificity was likely a combined effect of statistical power and biological difference. We further delved into the functions of different microbiota across different tissue-specific and tissue-sharing groups. Our tissue specificity and functional analysis together revealed the mechanistic

roles of specific regional intestinal tissues in regulating the composition of relevant dominant genera in the microbiome.

The crosstalk between the gut and other organs has been gradually uncovered and revealed to play key roles in mediating a variety of human diseases. However, the comprehensive regulatory framework involving intestinal gene expression, the gut microbiome, and disease loci has yet to be delineated. In this study, we integrated the established gene-to-microbiome regulatory maps with GWAS datasets to identify the disease loci that are dependent on the mechanism of the crosstalk between intestinal gene expression and microbiome. We conducted both heritability enrichment analyses and disease loci identification. We found that the gene-to-microbiome regulatory loci are enriched for the GWAS variants of multiple genetic diseases, including respiratory traits and diseases, immune-related diseases, and neurodegenerative disorders, providing microbiome-dependent mechanisms underlying the corresponding gut–organ axes.

At the locus level, we characterized hundreds of disease loci mediated by the gene-to-microbiome regulation across the three intestinal tissues. We unraveled two modes of pleiotropic effects in the regulatory axis formed by intestinal gene expression, microbiome composition, and disease. For the microbiome-level pleiotropy, we show an example where the colocalization between schizophrenia GWAS and *ATG13* is coupled with three different genera, including *Bifidobacterium*, *Ruminococcaceae UCG002*, and *Ruminococcaceae UCG009*. For the disease-level pleiotropy, we uncovered the regulatory effect of *CDK2AP1* expression on *Allisonella* regulatory pair that interprets the heritability of multiple traits, including asthma, allergic rhinitis, and FEV. Of note, while we accounted for multiple testing by *P* value corrections for the majority of our analyses, it was technically difficult for the multi-trait colocalization analysis.

Our study has several limitations. Since the identification of gene-to-microbiome regulation is dependent on the genetic maps of the gene expression and microbiome, the statistical power of these maps may affect the sensitivity of the analysis. While the GTEx-based eQTLs are well-powered, the currently available mbQTLs are of relatively insufficient power. Based on the estimation from population cohorts (Qin et al, 2022), the majority of taxa are expected to be present in fewer than 50% of the samples, which would cause a reduction of the effective sample size during the mbQTL identification (Lopera-Maya et al, 2022). We envision that more comprehensive bidirectional regulatory maps between gene expression and microbiome can be established with enhanced mbQTL datasets in the future. Moreover, since the eQTLs and mbQTLs were collected from two separate cohorts, we utilized a two-sample MR framework to mitigate potential individual-level confounders. Future studies focusing on a single cohort with a comprehensive curation of phenotypes can help disentangle gene-to-microbiome regulation from potential confounders. Third, the GTEx eQTLs are measured at the bulk tissue level and therefore limit our understanding of the contribution of specific cell types in regulating the microbiome composition. This issue will likely be tackled with the rapidly increasing single-cell RNA-seq datasets available. Fourth, the current analyses were restricted to European ancestry due to QTL availability. Cross-ancestry studies may provide a more comprehensive understanding of the ancestry specificity of gene–microbiome crosstalk. Lastly, the current study is confined to the regulation between gene expression and microbiome, without in-depth investigation into other molecular layers. Since studies have reported

that microbiota can affect the epigenome of the host, it will be of great interest to systematically interrogate the multi-layer crosstalk between the microbiome and host genetics across gene expression, proteomics, and metabolomics.

Overall, our multi-tissue and multi-layer regulatory paths across intestinal gene expression, the microbiome, and human complex disease provide a molecular foundation for the regulatory roles of the microbiome in human health and diseases. The identification of pathway enrichments of gene-to-microbiome regulatory pairs and the microbiome-dependent disease loci together deepens our understanding of the gut–organ axes and paves the way for potential therapeutics targeting the microbiome.

# Methods

### Reagents and tools table

| Reagent/resource | Reference or source | Identifier or catalog number |
| --- | --- | --- |
| **Experimental models** | | |
| Chronic restraint stress mice | Zhejiang Vital River | 219 |
| Mice | Zhejiang Vital River | 219 |
| *Allisonella histaminiformans* | Culture Collection University of Gothenburg (CCUG) | CCUG 48567 T |
| *Parasutterella muris* | Guangdong Microbial Culture Collection Center | GDMCC 1.4044 |
| *Blautia obeum* | Guangdong Microbial Culture Collection Center | GDMCC 1.5591 |
| *Dorea longicatena* | Culture Collection University of Gothenburg (CCUG) | CCUG 45247 |
| **Chemicals, enzymes, and other reagents** | | |
| Total Bile Acid (TBA) Content Assay Kit | Solarbio | #BC6300 |
| Acetate Assay Kit | Abcam | #ab204719 |
| Mice Short-chain Fatty Acids (SCFAs) Kit | MEIMIAN | #MM-0994M1 |
| DSMZ1006 medium | Hunan Weiway Mirazyme Biotechnology Co., Ltd. | #DM1006 |
| BHI medium | Becton, Dickinson and Company | #BD 237500 |
| DSMZ110 medium | Hunan Weiway Mirazyme Biotechnology Co., Ltd. | #DM110 |
| DSMZ78 medium | Hunan Weiway Mirazyme Biotechnology Co., Ltd. | #DM78 |
| **Software** | | |
| TwoSampleMR (v0.5.6) | Hemani et al, 2018 | |
| coloc (v.5.2.2) | Giambartolomei et al, 2014 | |
| clusterProfiler | Yu et al, 2012 | |
| moloc (v0.1.0) | Giambartolomei et al, 2018 | |
| DOSE | Yu et al, 2015 | |
| S-LDSC (v1.0.1) | Bulik-Sullivan et al, 2015 | |
| SMR | Zhu et al, 2016 | |

# Methods and protocols

### Collection of eQTL, mQTL, and mbQTL datasets

To identify the gene expression-to-microbiome regulation in the intestinal tissues, we collected the summary statistics of eQTLs for transverse colon, sigmoid colon, and ileum from the GTEx consortium, and gut mbQTLs consisting of 116 microbial genera from the MiBioGen (Kurilshikov et al, 2021; Data ref: Kurilshikov et al, 2021; GTEx Consortium, 2020; Data ref: GTEx Consortium, 2020). The three intestinal eQTL datasets are all from European ancestry, and the sample size of each is 368, 318, and 174, respectively. The sample size for mbQTLs is 18,340, and these datasets mainly come from European ancestry. We collected the mQTLs for the transverse colon only from the enhancing GTEx (eGTEx), as mQTL datasets for the other two intestinal tissues were unavailable. The dataset consists of the individuals of European ancestry with a sample size of 987 (Oliva et al, 2023; Data ref: Oliva et al, 2023). The detailed information for the datasets is available in Table EV1.

### Mendelian randomization

We performed Mendelian randomization (MR) to detect the regulatory relationship from intestinal gene expression (as exposure) to microbiome composition (as outcome) across the three tissues, including the transverse colon, sigmoid colon, and ileum. The inverse variance weighted (MR-IVW) analysis was carried out using the R package TwoSampleMR (v0.5.6) (Hemani et al, 2018). For instrumental variables (IVs), we selected significant eQTLs defined by GTEx that were after two rounds of multiple testing corrections (GTEx Consortium, 2020), and further applied a $P$ value threshold of $5 \times 10^{-8}$ to increase the reliability of the MR analysis. Subsequent LD clumping with a strict threshold ($r^2 < 0.01$ within 100 kb windows) was conducted to pick out independent instruments using 1000 Genomes Project phase 3 as the reference panel (1000 Genomes Project Consortium et al, 2015). To test if the microbiome composition was linked to the intestinal gene expression and DNA methylation sites, we further conducted MR analyses to identify the microbiome-to-gene expression and microbiome-to-methylation regulatory pairs. Significant mbQTLs with an association $P$ value $< 1 \times 10^{-5}$ were selected as instrumental variables. Multiple testing corrections were implemented for the nominal MR $P$ values using the study-wise FDR correction method, and FDR threshold was set to 0.05. To ensure the reliability of the analysis, we excluded the MR pairs inferred from single IVs and only retained those obtained from multiple IVs. In addition, we utilized the MR-Egger test to exclude those pairs showing potential pleiotropic effects ($P$ value $< 0.05$). We further conducted the leave-one-out analysis to assess whether any causal effect estimates are driven by one SNP independently. To investigate the regulatory relationship from gene-to-microbiome regulation to diseases or traits, we applied two-step MR analysis, which refers to doing MR analysis sequentially, where the outcome of the first MR serves as the exposure variable in the subsequent MR analysis. We conducted the heterogeneity in dependent instruments (HEIDI) test to determine whether the causal relationship from gene expression to microbial abundance was driven by a shared genetic variant or a linkage model. MR results with a significant $P_{HEIDI}$ value under 0.01 were filtered out for further analysis (Zhu et al, 2016).

### Classification of the distinct two regulatory patterns

We implemented hierarchical clustering of MR regulatory pairs for each of the three intestinal tissues and categorized the genes into

two distinct groups based on the number of microbiota they regulated: one group contained genes that broadly regulate the microbiome (genera count >10), while the other consisted of genes that more specifically regulate the microbiome (genera count <10).

### Colocalization analysis

We conducted colocalization analysis for all the MR pairs tested for gene expression-to-microbiome, microbiome-to-gene expression, and microbiome-to-DNA methylation using the R package coloc v.5.2.2 (Giambartolomei et al, 2014). Coloc returns posterior probabilities indicating the likelihood that the following scenarios are true: there is no association at the locus with either gene expression or the abundance of microbiota (PP0); there is an association with gene expression but not microbial abundance (PP1); there is no association with gene expression but there is an association with microbial abundance (PP2); there is an association with both the gene expression and the microbiome but with distinct causal variants (PP3); there is an association with both the gene expression and the microbial abundance with a shared causal variant (PP4). We considered PP4 >0.5 as robust evidence of colocalization between eQTLs and mbQTLs, as well as between mbQTLs and mQTLs.

### Tissue specificity of MR regulatory pairs

The MR pairs of three tissues were divided into seven categories based on the sharing across the three tissues, including three tissue-specific, three pairwise sharing, and one all-sharing group, and the Venn diagram was plotted to display the intersection of regulatory pairs determined for each category. In addition, we calculated the intersection of eGenes (genes with eQTLs) across three tissues. For the tissue-specific groups of MR regulatory pairs, we further classified them as follows: MR pairs containing loci present in the tissue-shared groups of eQTLs were defined as the gene expression-to-microbiome-specific subgroup, and MR pairs with loci exclusively found in the tissue-specific groups of eQTLs were categorized as the eQTL-specific subgroup.

### Consistency of effect sizes in MR regulatory pairs across the three tissues

We compared the effect sizes and the directionalities of the gene expression-to-microbiome composition regulatory loci. Significant MR regulatory pairs were selected (FDR < 0.05) in three tissues, respectively, and pairwise Pearson correlation coefficients were estimated using the weighted regression. The directionality consistency across three tissues was calculated by comparing the regulatory pairs with consistent directions to all pairs derived from the gene expression-to-microbiome MR pairs used for the comparisons.

### Enrichment of microbiome in tissue-specific and tissue-sharing groups

The enrichment of each microbiome in tissue-specific and tissue-shared regulatory pairs was quantified by odd ratio using the following equation: $OR = (n/(n_s - n))/((N - n)/(N_s - N - (n_s - n)))$, where $n$ represents the number of genetic loci involved in regulating a specific microbiome within one of the tissue-specific and tissue-sharing regulation categories, $n_s$ indicates the total number of genetic loci in a category that regulate all microbiota, $N$ denotes the total number of genetic loci regulating one microbiome

across all categories, and $N_s$ is the total number of regulatory pairs governing all microbiota across all categories. The $P$ value for enrichment was determined using a right-sided Fisher's exact test.

### Functional annotation

We applied R package clusterProfiler to perform Gene Ontology (GO) analyses based on the database Gene Ontology (Ashburner et al, 2000; Yu et al, 2012). Gene sets used for GO enrichment in the biological process term were genes from "broadly regulating" group, "specially regulating" group, and tissue-specific and tissue-sharing regulatory groups. In addition, to understand the potential disease implications for the tissue specificity of the regulation, Disease Ontology (DO) enrichment analysis was performed via the DOSE package based on the Disease Ontology database for the genes involved in the seven gene expression-to-microbiome regulatory categories (Ashburner et al, 2000; Baron et al, 2024; Yu et al, 2015). For the GO enrichment analysis of the genes involved in the "broadly-regulating" and "specifically-regulating" across three tissues, all genes tested in the MR analysis for each respective tissue were used as the background set, regardless of significance. For the GO and DO enrichments of the genes in the different tissue-sharing and tissue-specific groups, the background gene list consisted of all genes identified as significant MR pairs across the three tissues. In both GO and DO analyses, $P$ value < 0.05 was used as a threshold to select the significant enrichment GO terms.

### GWAS summary statistics collection

We obtained the disease/trait GWAS datasets from GWAS ATLAS, Open GWAS, and UK Biobank for the multi-trait colocalization and the heritability enrichment analysis (Watanabe et al, 2019; Data ref: Watanabe et al, 2019; Elsworth et al, 2020; Data ref: Elsworth et al, 2020; Sudlow et al, 2015; Data ref: Sudlow et al, 2015) (Dataset EV7). As for the GWAS data from GWAS ATLAS and OpenGWAS, we imputed the missing MAF using the 1000 Genomes Project phase 3 as reference. For the UK Biobank data, we calculated MAF as a weighted average of allele frequencies in cases and controls, accounting for the proportion of cases and controls in the European population. The rsIDs in all the GWAS datasets were lifted over to human genome build version 38 (GRCh38/hg38). We standardized the effect allele and the other allele across all the collected GWAS datasets according to the QTL datasets, and correspondingly reversed the signs of effect sizes for any loci that were inconsistent.

### GWAS heritability partition of gene expression-to-microbiome

The LD score regression (LDSC) approach was utilized to analyze the enrichment of gene-to-microbiome regulation in 89 disease/trait GWAS datasets (Finucane et al, 2015; Bulik-Sullivan et al, 2015). We used one baseline model v1.1 for the calculation of the heritability enrichment. For the significant regulatory pairs in the three tissues, we aggregated the genetic loci regulating all microbiota in each tissue and used these loci for heritability analysis. We applied the enrichment value >1 and enrichment $P$ value < 0.05 as a threshold to select the significant enrichment.

### Multi-trait colocalization analysis

We conducted Bayesian-based multi-trait colocalization analyses to detect the variants associated with gene expression, microbiome,

and diseases/traits using the R package moloc (v0.1.0) (Giambar-tolomei et al, 2018), across eQTLs of three tissues, mbQTLs, and diseases/traits GWAS. Moloc computed the posterior probabilities for 15 possible configurations of the associations. A threshold of 0.1 for the posterior probability colocalization (PP abc) across eQTL, mbQTL, and GWAS was utilized. We also conducted multi-trait colocalization analysis to identify the regulatory path across microbiome, DNA methylation, and gene expression, following the same procedure and filtering threshold described above.

### Chronic restraint stress (CRS) model construction

Over a period of three weeks, mice were subject to confinement in well-ventilated 50 mL centrifuge tubes for 3 h each day, at random times in the morning or afternoon. During the restraint period, the CRS mice were unable to move freely within the tubes. To control the potential impacts of food and water deprivation during restraint, controlled mice were similarly deprived of food and water for the same duration as the CRS mice, but without being physically restrained (Sheng et al, 2025). All animal studies were performed in compliance with the guidelines for the care and use of laboratory animals, and the protocol was approved by the Medical Experimental Animal Care Commission of Zhejiang University (No. ZJU20240311).

### Mouse colonization with bacteria

The bacteria used in this study were cultured to the appropriate optical density (OD) value or to the stationary phase. A volume of 12 mL of the bacterial suspension was centrifuged at 8000 rpm for 10 min to obtain the bacterial pellet. The pellet was then resuspended in 1 mL of the original culture medium to create a uniform suspension. Wild-type (WT) male mice were orally administered 200 µL of the microbial suspension daily for 5 consecutive days. Meanwhile, mice in the control group received the equivalent volume of the original culture medium via oral gavage. Each group contains five mice. The abundance of gut microbiota was detected by the primers shown in the Dataset EV9.

### Detection of mouse serum metabolites

Blood was collected from mice and centrifuged at 3000 rpm for 10 min to obtain hemolysis-free serum. According to the manu-facturer's instructions, the total bile acid (TBA) Content Assay Kit (Solarbio, BC6300) was used to measure the TBA content in the mouse serum. The acetate concentration in the serum was measured using the Acetate Assay Kit (Abcam, ab204719). The total short-chain fatty acid (SCFA) content in the serum was determined using the Mice SCFAs Kit (MEIMIAN, MM-0994M1).

### Microbial strains and culture conditions

All bacteria were anaerobic and cultured under strict anaerobic conditions at 37 °C in an anaerobic chamber. *Allisonella histaminifor-mans* CCUG 48567 T was obtained from the Culture Collection University of Gothenburg (CCUG) and cultured in DSMZ1006 medium. *Parasutterella muris* GDMCC 1.4044 was obtained from the Guangdong Microbial Culture Collection Center and cultured in BHI medium (BD 237500). *Blautia obeum* GDMCC 1.5591 was also obtained from the Guangdong Microbial Culture Collection Center and cultured in DSMZ110 medium. *Dorea longicatena* CCUG 45247 was obtained from the Culture Collection University of Gothenburg (CCUG) and cultured in DSMZ78 medium.

## Data availability

All the summary statistics data of eQTLs, mbQTLs, mQTLs, and GWAS are available publicly, with detailed information available in Table EV1 and Dataset EV7. Code for the data processing and statistical genetics analysis is accessible at the GitHub repository (https://github.com/xiongxslab/microbiomeMR).

The source data of this paper are collected in the following database record: biostudies:S-SCDT-10_1038-S44320-025-00173-7.

## Peer review information

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

## Acknowledgements

We thank the members of the Xiong Lab and Sheng lab for their discussion and suggestions throughout the project. We thank the support from the core facilities and computing platform of Liangzhu Laboratory at Zhejiang University. This work was supported by the Ministry of Science and Technology of China (nos. 2023YFA1800700 and 2024YFF1207600 to XX), Fundamental Research Funds for the Central Universities (no. 226-2025-00176 to XX), National Natural Science Foundation of China (nos. 32422017, 32370609 and 92353301 to XX), the Natural Science Foundation of Zhejiang Province (nos. LR25C060002 to XX), the Zhejiang Provincial Leading Innovation and Entrepreneurship Team Introduction and Cultivation Program (no. 2024R01024 to XX), and the funding from the State Key Laboratory of Transvascular Implantation Devices (no. 012024002 to XX), Benyuan Foundation, and K.C. Wong Education Foundation to XX.

## Author contributions

**Haochuan Wang**: Data curation; Formal analysis; Validation; Visualization; Methodology; Writing—original draft; Writing—review and editing. **Chengyu Li**: Data curation; Formal analysis; Validation; Visualization; Methodology; Writing—original draft; Writing—review and editing. **Zhen Hu**: Validation; Methodology. **Haonan Feng**: Methodology. **Luowei Chen**: Methodology. **Ke Ding**: Methodology. **Jiuhong Nan**: Methodology. **Yuhan Wu**: Formal analysis; Methodology. **Jinghao Sheng**: Supervision; Methodology; Writing—review and editing. **Xushen Xiong**: Supervision; Funding acquisition; Investigation; Writing—original draft; Writing—review and editing.

Source data underlying figure panels in this paper may have individual authorship assigned. Where available, figure panel/source data authorship is listed in the following database record: biostudies:S-SCDT-10_1038-S44320-025-00173-7.

## Disclosure and competing interests statement

The authors declare no competing interests.

# Expanded View Figures

**Figure EV1.  Genetic regulatory maps between gene expression and microbiome.**

(**A**) Manhattan plot illustrating the significant gene expression-to-microbiome regulatory pairs in the sigmoid colon and ileum. Each dot represents the gene, with the *x* axis indicating the position and the *y* axis means the $-\log_{10} P$ value. (**B**) Effect size consistency the MR regulatory pairs identified based on GTEx (discovery) versus CEDAR (replication) datasets. Pearson correlation coefficient and the corresponding *P* value were calculated. (**C**) Distribution of the significant gene-to-microbiome regulatory pairs in each genus across three tissues. The box's central line indicates the median. The bounds of the box represent the 25th and 75th percentiles (interquartile range, IQR). The whiskers extend to the most extreme data points that are within 1.5 × IQR from the box. The value of n shown in each *x* axis label represents the number of gene-to-microbiome regulatory pairs used for that genus. (**D**) The effect sizes of the gene-to-microbiome regulation of the top 20 genera in the sigmoid colon and ileum. The *y* axis indicates the absolute effect size, and the *x* axis indicates the top 20 genera with the strongest overall regulatory magnitude. The box's central line indicates the median. The bounds of the box represent the 25th and 75th percentiles (interquartile range, IQR). The whiskers extend to the most extreme data points that are within 1.5 × IQR from the box. The value of n shown in each *x* axis label represents the number of gene-to-microbiome regulatory pairs used for that genus. (**E**) Box plot indicating the functional annotation of the genes from the regulatory MR pairs on chromosome 12 in three tissues. (**F**) Number of MR-based microbiome-to-gene pairs, colocalization pairs across three tissues. (**G**) Number of MR-based microbiome-to-methylation pairs, colocalization pairs in the transverse colon. Source data are available online for this figure.

▶

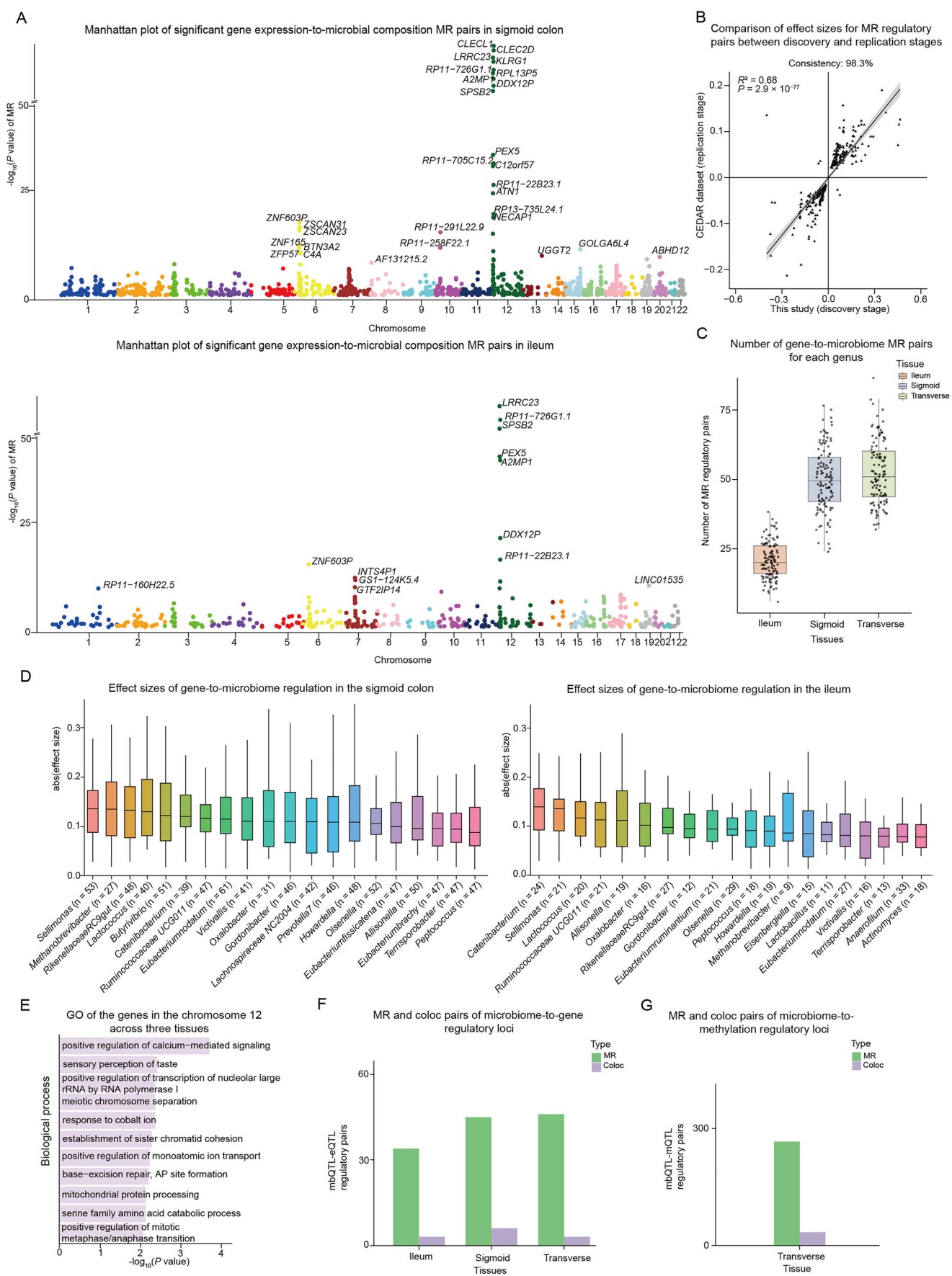

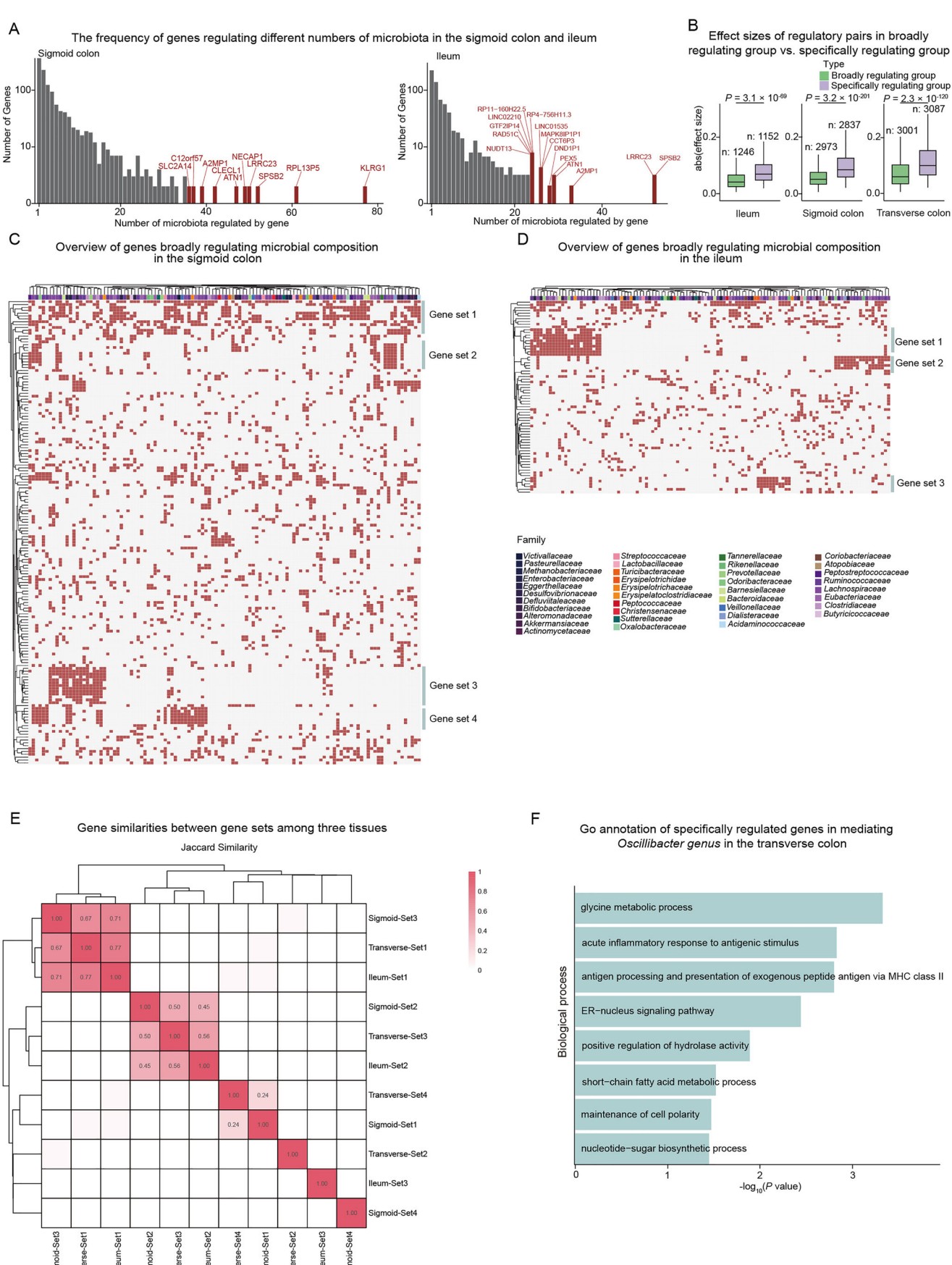

**Figure EV2.  Functional enrichments of gene-to-microbiome regulatory loci.**

(A) Bar plot indicating the distribution of the number of genes based on the number of microbiota they regulated in the sigmoid colon and ileum. Genes regulating more than 50 microbiota genera were labeled. (B) Comparison of the overall effect sizes of gene-to-microbiome pairs between broadly regulating groups and specifically regulating groups in the three tissues. The box's central line indicates the median. The bounds of the box represent the 25th and 75th percentiles (interquartile range, IQR). The whiskers extend to the most extreme data points that are within 1.5 × IQR from the box. The value of n shown above each box represents the number of gene-to-microbiome regulatory pairs in each group. (C) Heatmap showing the genes that broadly regulated microbiota in the sigmoid colon, with five and three gene sets formed via hierarchical clustering. (D) Heatmap showing the genes that broadly regulated microbiota in the ileum, with five and three gene sets formed via hierarchical clustering. (E) Heatmap indicating the similarity level (Jaccard similarity index) between gene sets across the three tissues. The Jaccard index was measured by the presence-absence gene matrix for three tissues. (F) An example of the GO enrichment for the genes that specifically regulated the *Oscillibacter* genus. Functional enrichments were calculated by using Fisher's exact test. Source data are available online for this figure.

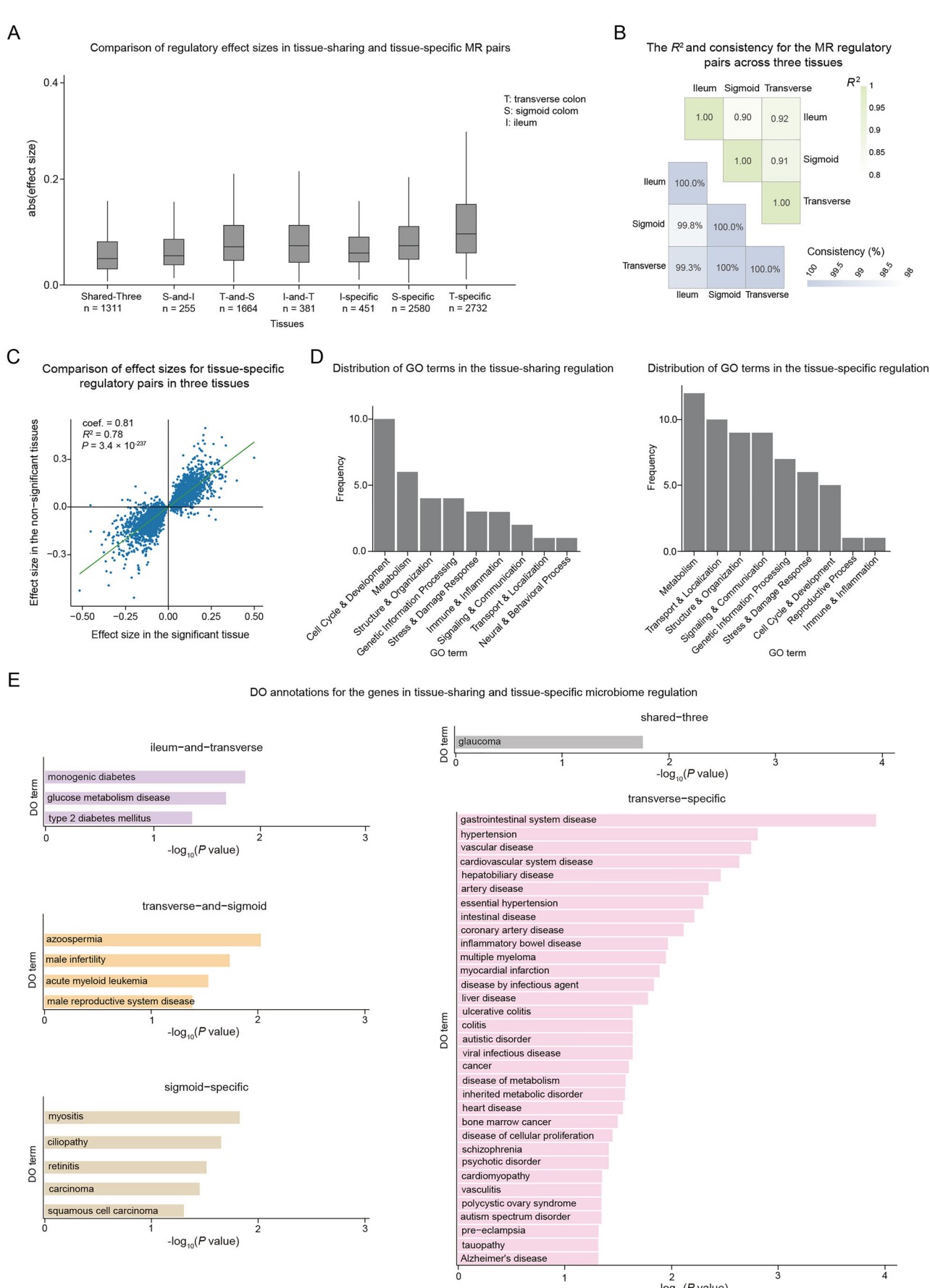

**A** Comparison of regulatory effect sizes in tissue-sharing and tissue-specific MR pairs

T: transverse colon
S: sigmoid colom
I: ileum

**B** The $R^2$ and consistency for the MR regulatory pairs across three tissues

**C** Comparison of effect sizes for tissue-specific regulatory pairs in three tissues

coef. = 0.81
$R^2 = 0.78$
$P = 3.4 \times 10^{-237}$

**D** Distribution of GO terms in the tissue-sharing regulation

Distribution of GO terms in the tissue-specific regulation

**E** DO annotations for the genes in tissue-sharing and tissue-specific microbiome regulation

◀ **Figure EV3.  Tissue specificity and sharing of the gene-to-microbiome regulation.**

(**A**) Comparison of the effect sizes in the tissue-sharing and tissue-specific groups, with *x* axis illustrating the MR regulatory types, and *y* axis showing the absolute value of the effect size. The box's central line indicates the median. The bounds of the box represent the 25th and 75th percentiles (interquartile range, IQR). The whiskers extend to the most extreme data points that are within 1.5 × IQR from the box. The value of *n* shown beneath each *x* axis label represents the number of gene-to-microbiome regulatory pairs in each group. (**B**) Heatmap illustrating the $R^2$ (upper triangle) and consistency (lower triangle) for the regulatory pairs across three intestinal tissues. (**C**) Comparisons of the effect sizes for the tissue-specific regulatory pairs between the significant tissue versus other tissues. Regression coefficient, Pearson correlation coefficients, and the corresponding *P* values were calculated. (**D**) Bar plots indicating GO term distributions in the tissue-sharing and tissue-specific regulation. (**E**) Bar plots indicating the DO enrichment for the genes involved in the tissue-sharing and tissue-specific regulation categories. DO enrichments were calculated by using Fisher's exact test. Source data are available online for this figure.

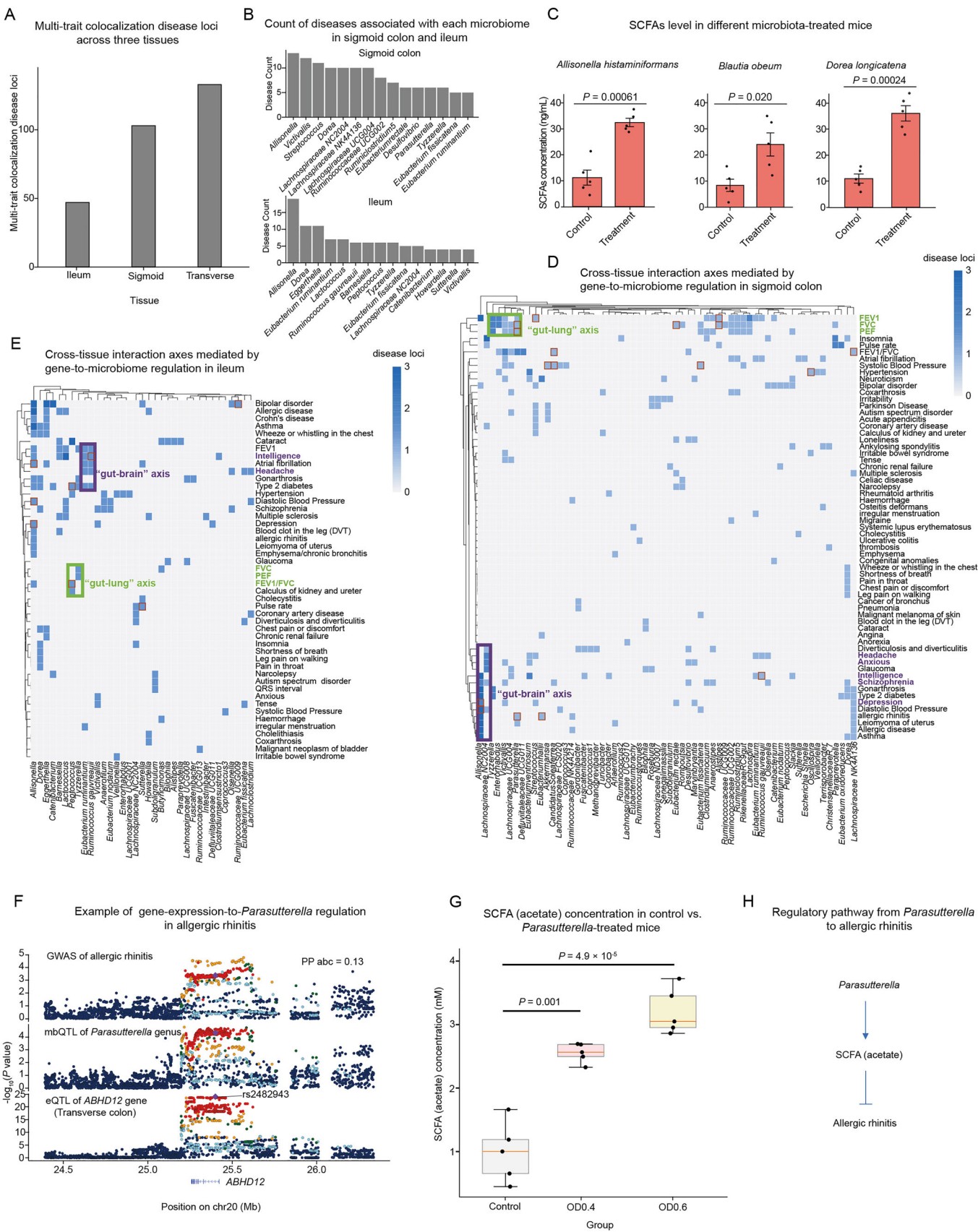

◀ **Figure EV4.   Gene-to-microbiome regulation underlying genetic diseases.**

(A) Bar plot indicating the number of multi-trait colocalization disease loci across three tissues. (B) Bar plots illustrating the number of genetic diseases and traits (*y* axis) associated with different microbiota (*x* axis) identified through multi-trait colocalization analysis in the sigmoid colon and ileum. (C) Experimental validations for the regulatory effects of different microbiota on (short-chain fatty acid) SCFA production. *P* value for each comparison was calculated by unpaired *t* test. Each group contained 5 replicates. Error bars indicate standard error of the mean. (D) Heat maps indicating the multi-trait colocalization between complex diseases (row) and the genera (column) involved in gene-to-microbiome regulation pairs in the sigmoid colon, with the color scale representing the number of loci. The red boxes represent the loci that were further supported by two-step MR analyses. (E) Heat maps indicating the multi-trait colocalization between complex diseases (row) and the genera (column) involved in gene-to-microbiome regulation pairs in the ileum, with the color scale representing the number of loci. The red boxes represent the loci that were further supported by two-step MR analyses. (F) LocusCompare visualization of the multi-trait localization across allergic rhinitis GWAS, *Parasutterella* genus mbQTL, and *ABHD12* eQTL in the transverse colon. The *y* axis shows log-changed *P* value of association tests in GWAS, mbQTLs, and eQTLs. The lead variant is indicated, and the correlation of the LD effect relative to the lead variant is represented by the color scale. (G) Box plot representing comparisons of the SCFA (acetate) concentration between control mice and mice treated with different concentrations of *Parasutterella*. The box's central line indicates the median. The bounds of the box represent the 25th and 75th percentiles (interquartile range, IQR). The whiskers extend to the most extreme data points that are within 1.5 × IQR from the box. *P* value for each comparison was calculated by unpaired *t* test. Each group contains 5 replicates. (H) Schematic illustrating the regulatory axis across *Parasutterella* abundance, SCFA level, and allergic rhinitis. Source data are available online for this figure.

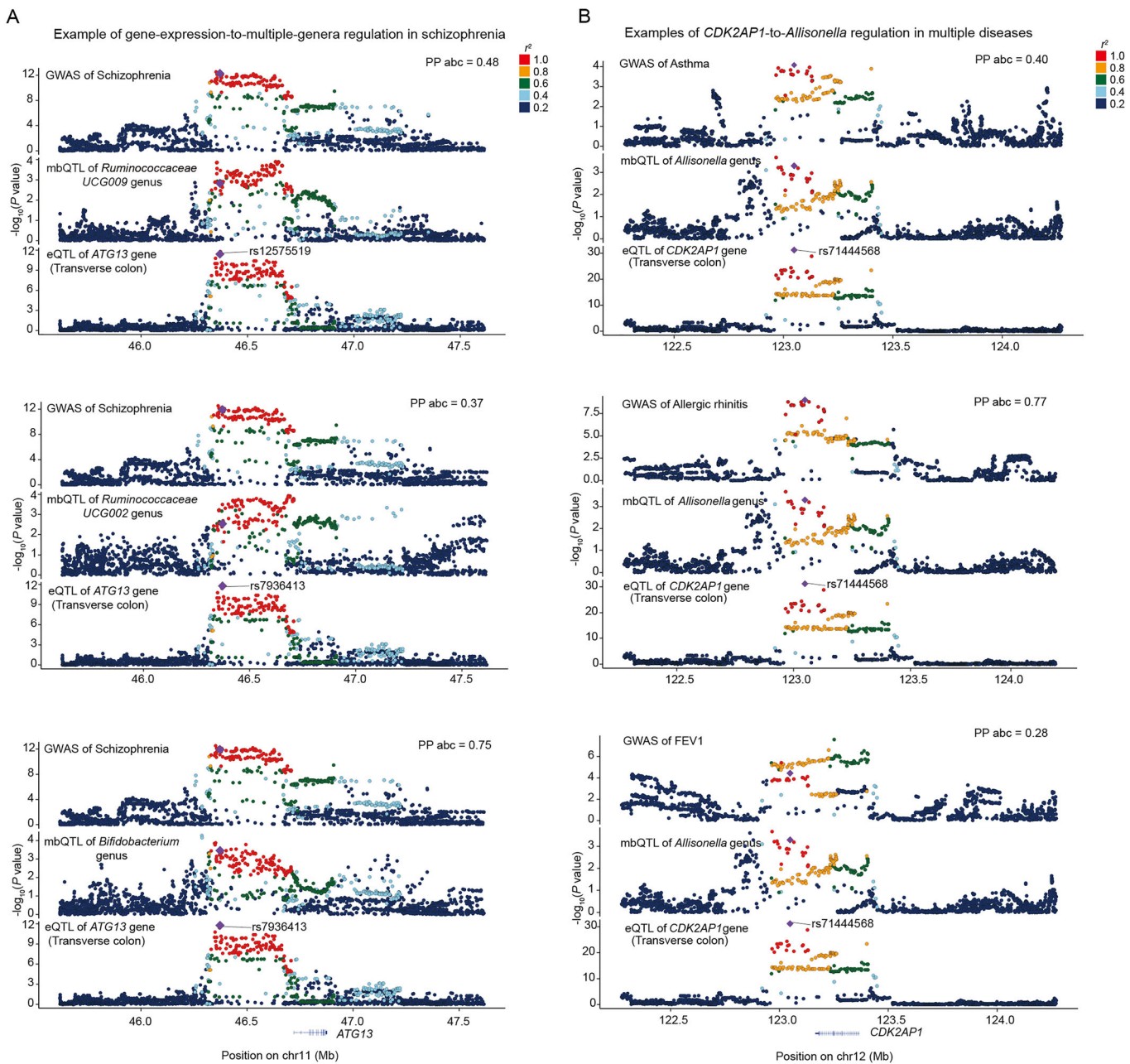

**Figure EV5.   Pleiotropic effect of gene-to-microbiome regulation on the diseases.**

(**A**) LocusCompare plot of a pleiotropic disease locus mediated by various microbial genera, including *Ruminococcaceae UCG009* (top), *Ruminococcaceae UCG002* (middle), *and Bifidobacterium* (bottom), for their colocalization with schizophrenia GWAS and *ATG13* locus in the transverse colon. The lead variant and the posterior probability for each multi-trait colocalization are shown. (**B**) LocusCompare plots showing *CDK2AP1-to-Allisonella regulation* is associated with various diseases, including asthma, allergic rhinitis and FEV in the transverse colon, the lead variant and the posterior probability for each multi-trait colocalization are shown. Source data are available online for this figure.

