## [Peer Review File · Molecular Systems Biology]

Genetics-mediated regulation of intestinal gene expression on microbiome contributes to human disease heritability

Haochuan Wang, Chengyu Li, Zhen Hu, Haonan Feng, Luowei Chen, Ke Ding, JiuHong Nan, Yuhan Wu, Jinghao Sheng, and Xushen Xiong

Corresponding author(s): Xushen Xiong (xiongx@zju.edu.cn) , Jinghao Sheng (jhsheng@zju.edu.cn)

Review Timeline:

Submission Date:	20th Jun 25
Editorial Decision:	25th Jul 25
Revision Received:	15th Oct 25
Editorial Decision:	13th Nov 25
Revision Received:	17th Nov 25
Accepted:	20th Nov 25

Editor: Jingyi Hou

Transaction Report:

25th Jul 2025

Manuscript Number: MSB-2025-13191

Title: Genetics-mediated regulation of intestinal gene expression on microbiome contributes to human disease heritability

Author: Haochuan Wang

Chengyu Li

Ke Ding

Jiuhong Nan

Luwei Chen

Yuhan Wu

Xushen Xiong

Dear Prof Xiong,

Thank you for submitting your work to Molecular Systems Biology. We have now heard back from the three reviewers who agreed to evaluate your manuscript. As you will see from the reports below, the reviewers acknowledge the interest of the study. They raise, however, a series of concerns, which we would ask you to address in a major revision.

Without reiterating all the points raised in the reviews below, we highlight the following substantial issues that warrant attention:

- Concerns regarding statistical methods, threshold selection, and validation;
- Reviewer #2's comment about the lack of control for individual-level confounding factors;
- Reviewer #3's concerns about the correlative nature of the findings and the absence of mechanistic insight.

During the pre-decision cross-commenting phase, Reviewers #2 and #3 provided additional constructive feedback, which is included following the main reviewer reports. We encourage you to carefully address the concerns in light of these additional comments.

All other issues need to be addressed as well. As you may already know, our editorial policy allows in principle a single round of major revision, and it is therefore essential to provide responses to the reviewers' comments that are as complete as possible. Please feel free to contact me in case you would like to discuss in further detail any of the issues raised by the reviewers.

On a more editorial level, we would ask you to address the following issues:

- Please provide a .docx formatted version of the manuscript text (including legends for main figures, EV figures and tables). Please make sure that the changes are highlighted to be clearly visible.
- Please provide individual production quality figure files as .eps, .tif, .jpg (one file per figure).
- Please provide a .docx formatted letter INCLUDING the reviewers' reports and your detailed point-by-point responses to their comments. As part of the EMBO Press transparent editorial process, the point-by-point response is part of the Review Process File (RPF), which will be published alongside your paper.
- Please note that all corresponding authors are required to supply an ORCID ID for their name upon submission of a revised manuscript.
- We replaced Supplementary Information with Expanded View (EV) Figures and Tables that are collapsible/expandable online (see examples in <http://msb.embopress.org/content/11/6/812>). A maximum of 5 EV Figures can be typeset. EV Figures should be cited as 'Figure EV1, Figure EV2' etc... in the text and their respective legends should be included in the main text after the legends of regular figures.

Additional Tables/Datasets should be labeled and referred to as Table EV1, Dataset EV1, etc. Legends have to be provided in a separate tab in case of .xls files. Alternatively, the legend can be supplied as a separate text file (README) and zipped together with the Table/Dataset file.

For the figures and tables that you do NOT wish to display as Expanded View figures, they should be bundled together with their legends in a single PDF file called *Appendix*, which should start with a short Table of Content. Each legend should be below the corresponding Figure/Table in the Appendix. Appendix figures and tables should be referred to in the main text as: "Appendix Figure S1, Appendix Figure S2, Appendix Table S1" etc. See detailed instructions regarding expanded view here: <https://www.embopress.org/page/journal/17444292/authorguide#expandedview>.

- Before submitting your revision, primary datasets (and computer code, where appropriate) produced in this study need to be

deposited in an appropriate public database (see <http://msb.embopress.org/authorguide> - [dataavailability](https://www.embopress.org/page/journal/17444292/authorguide#dataavailability) <https://www.embopress.org/page/journal/17444292/authorguide#dataavailability>).

The accession numbers and database should be listed in a formal "Data Availability" section (placed after Materials & Method) that follows the model below (see also <https://www.embopress.org/page/journal/17444292/authorguide#dataavailability>). Please note that the Data Availability Section is restricted to new primary data that are part of this study.

Data availability

- RNA-Seq data: Gene Expression Omnibus GSE46843 (<https://www.ncbi.nlm.nih.gov/geo/query/acc.cgi?acc=GSE46843>)

- [data type]: [name of the resource] [accession number/identifier/doi] ([URL or identifiers.org/DATABASE:ACCESSION])

Additional information on source data and instruction on how to label the files are available

- Our journal encourages inclusion of *data citations in the reference list* to directly cite datasets that were re-used and obtained from public databases. Data citations in the article text are distinct from normal bibliographical citations and should directly link to the database records from which the data can be accessed. In the main text, data citations are formatted as follows: "Data ref: Smith et al, 2001". In the Reference list, data citations must be labeled with "[DATASET]". A data reference must provide the database name, accession number/identifiers and a resolvable link to the landing page from which the data can be accessed at the end of the reference. Further instructions are available at .

- We updated our journal's competing interests policy in January 2022 and request authors to consider both actual and perceived competing interests. Please review the policy <https://www.embopress.org/competing-interests> and update your competing interests if necessary.

Please use the heading "Disclosure statement and competing interests".

- All Materials and Methods need to be described in the main text using our 'Structured Methods' format. According to this format, the Methods section includes a Reagents and Tools Table (listing key reagents, experimental models, software and relevant equipment and including their sources and relevant identifiers) followed by a Methods and Protocols section describing the methods, ideally using a step-by-step protocol format. The aim is to facilitate adoption of the methodologies across labs. Please download and fill our Reagents and Tools Table template (.docx), which you can find in our author guidelines: <https://www.embopress.org/page/journal/17444292/authorguide#structuredmethods>.

-Regarding data quantification:

Please ensure to specify the name of the statistical test used to generate error bars and P values, the number (n) of independent experiments (please specify technical or biological replicates) underlying each data point and the test used to calculate p-values in each figure legend. Discussion of statistical methodology can be reported in the materials and methods section, but figure legends should contain a basic description of n, P and the test applied.

Graphs must include a description of the bars and the error bars (s.d., s.e.m.).

- Please provide a "standfirst text" summarizing the study in one or two sentences (approximately 250 characters, including space), three to four "bullet points" highlighting the main findings and a "synopsis image" (550px width and 400-600 px height, PNG format) to highlight the paper on our homepage.

Here are a couple of examples:

<https://www.embopress.org/doi/10.15252/msb.20199356>

<https://www.embopress.org/doi/10.15252/msb.20209475>

<https://www.embopress.org/doi/10.15252/msb.209495>

When you resubmit your manuscript, please download our CHECKLIST (<https://www.embopress.org/pb-assets/embosite/EMBO%20Press%20Author%20Checklist-1642513524327.xlsx>) and include the completed form in your submission.

Please note that the Author Checklist will be published alongside the paper as part of the transparent process (<https://www.embopress.org/page/journal/17444292/authorguide#transparentprocess>).

If you feel you can satisfactorily deal with these points and those listed by the referees, you may wish to submit a revised version

of your manuscript. Please attach a covering letter giving details of the way in which you have handled each of the points raised by the referees. A revised manuscript will be once again subject to review and you probably understand that we can give you no guarantee at this stage that the eventual outcome will be favorable.

I look forward to receiving your revised manuscript soon.

Kind regards,
Jingyi

Jingyi Hou, PhD
Senior Editor
Molecular Systems Biology

We realize that it is difficult to revise to a specific deadline. In the interest of protecting the conceptual advance provided by the work, we recommend a revision within 3 months (23rd Oct 2025). Please discuss the revision progress ahead of this time with the editor if you require more time to complete the revisions. Use the link below to submit your revision:

*** PLEASE NOTE *** As part of the EMBO Press transparent editorial process initiative (see our Editorial at <https://dx.doi.org/10.1038/msb.2010.72>), Molecular Systems Biology publishes online a Review Process File with each accepted manuscripts. This file will be published in conjunction with your paper and will include the anonymous referee reports, your point-by-point response and all pertinent correspondence relating to the manuscript. If you do NOT want this File to be published, please inform the editorial office at contact@molsystbiol.org within 14 days upon receipt of the present letter.

Reviewer #1:

This paper systematically explores a multi-tissue interaction map between intestinal gene expression and gut microbiome composition. By integrating with GWAS datasets, it identified the gene-microbiome-disease regulatory loci, shedding light on the microbiome-dependent mechanism for disease loci indicative of the gut-organ axes. In addition, the analyses were comprehensive and in-depth. I feel it is worth reading by more researchers in the related fields. Several comments are as following:

1. As mentioned in line 377-379, "The three intestinal eQTL datasets are all from European ancestry, and the sample size of each is 368, 318, and 174, respectively". Are these sample size sufficient to ensure valid and reliable estimates in two-sample Mendelian randomization?
2. Is the use of multiple-trait colocalization alone sufficient to infer that the microbiome mediates the effect of gene expression on disease? Additional evidence may be necessary to strengthen this conclusion.
3. Please ensure consistent formatting of figure captions figure captions throughout the manuscript (e.g., "Figure. 2A" in line 157, "Figure 2B" in line 160, and "Fig. 3g" in line 244).
4. The pronounced regulatory cluster on chromosome 12 is interesting. Could the authors explain its potential biological significance?
5. Please revise "Collection of eQTL, mQTL, and mQTL datasets" to "Collection of eQTL, mQTL, and mbQTL datasets".

Reviewer #2:

Overall Summary: Wang et al. investigate how host genetic regulation of intestinal gene expression influences gut microbiome composition and potentially contributes to complex human diseases. Using two-sample Mendelian Randomization and

colocalization analyses, the authors link eQTLs (from GTEx gut tissues) with microbiome QTLs (from MiBioGen) and 89 GWAS datasets. They construct tissue-specific gene-to-microbiome regulatory maps and identify immune- and metabolism-related pathways mediate gene-to-microbiome regulation across intestinal tissues. They found that gene-to-microbiome regulatory SNPs significantly contribute to the heritability of multiple complex diseases, particularly for respiratory and neuropsychiatric disorders, and identify microbial taxa such as Bifidobacterium and Ruminococcaceae that potentially mediate these effects.

This is a well written manuscript and addresses an important question in the field. However, there are a few concerns that need to be addressed:

Major critiques:

1. The study uses two entirely separate datasets (GTEx for eQTLs and MiBioGen for mbQTLs), which means gene expression and microbiome are never observed in the same individual. This prevents control for individual-level confounding, environmental exposures, such as medication or diet, which could independently influence both gene expression and the microbiome. Thus, the biological variation across tissues/cohorts may lead to misleading gene-microbiome regulatory associations.
2. How was the threshold of $PP4 > 0.5$ for colocalization determined? This value is relatively permissive and may not provide strong evidence for shared causal variants. It would be helpful to know whether any sensitivity analyses were conducted to assess the robustness of this threshold. Adopting a more stringent criterion or validating the findings in independent datasets would strengthen the evidence for shared genetic regulation.
3. The paper focuses on statistical significance, but rarely discusses effect sizes or confidence intervals which impedes the assessment of biological magnitude of gene-to-microbiome effects.
4. The functional interpretation of genera (e.g., SCFA production by *Allisonella* or Ruminococcaceae UCG003) relies on taxonomic annotations rather than direct functional or metabolomic data. Such genus-level inferences are speculative, as microbial functions are often strain-specific and context-dependent, requiring validation through metagenomics or metabolomics.
5. The tissue-specificity analysis compares tissue-specific vs. shared gene-to-microbiome regulatory pairs across intestinal sites, but does not fully control for the differing number of eQTLs, sample sizes, or microbiome MR pairs per tissue. As a result, the apparent tissue specificity could partly reflect differences in detection power rather than true biological differences in regulatory architecture.
6. What background genes were used for pathway enrichment analysis?

Minor critiques:

- In Fig 4C heatmap, abbreviations for traits (e.g., FVC, PEF) are used without immediately defining them in the figure or caption.
- In Fig 4D,E, the abbreviation "PP abc" is used without defining it in the main text or caption.

Reviewer #3:

Summary

The authors present a computational study aiming to characterize the genetic regulation of the gut microbiome via intestinal gene expression across three tissues (terminal ileum, transverse colon, and sigmoid colon). Leveraging Mendelian randomization (MR), colocalization, and methylation QTL data, they identify over 30,000 gene-microbiome regulatory pairs, some of which are further linked to disease heritability via GWAS loci and colocalization. The study proposes that tissue-specific and shared gene-microbiome interactions may contribute to inter-organ disease mechanisms such as the gut-lung and gut-brain axes.

Key conclusions include:

Thousands of host genes regulate gut microbiome genera in a tissue-specific or tissue-shared manner.

Certain genes (e.g., ATG13, A2M, RBM6) are implicated as microbiome-mediated contributors to disease loci via colocalization and trait enrichment.

Microbiome regulation may occur via gene expression and epigenetic modifications, and may influence systemic diseases through microbiome shifts.

The authors use GTEx-derived eQTLs from intestinal tissues, MiBioGen microbiome QTLs, methylation QTLs, and GWAS summary statistics. The main methodologies are two-sample MR, coloc, and multi-trait colocalization (moloc).

General Remarks

This is an ambitious manuscript that integrates several data modalities to investigate the genetic regulation of the gut microbiome and its relevance to human disease. The tissue-specific aspect and the inclusion of methylation data are valuable extensions to existing host-microbiome QTL analyses.

However the authors provide an extensive integrative analysis across multiple tissues and data types, the work lacks clear mechanistic insight and suffers from weak statistical rigor. The main findings are associative, not causal, and several methodological decisions (e.g., p-value thresholds, lack of validation) significantly weaken the robustness and interpretability of

the results. Particularly, the gene-microbiome-disease axes remain mostly speculative without functional support or deeper mechanistic insight. Additionally, several methodological choices require further justification or clarity. If the authors can address the concerns below-especially regarding statistical stringency, interpretation of disease relevance, and validation of key findings-this work would be significantly strengthened.

Major Concerns

1. While the manuscript reports thousands of associations between gene expression and microbiome features, it remains largely correlative in nature. There is no mechanistic model proposed, no clear causal chain established, and no experimentally testable hypothesis generated. The work functions more as a catalog than a mechanistic study.
2. The manuscript claims relevance to disease via GWAS integration, but the connections are shallow: No disease mechanism is unpacked in depth. No follow-up analysis, validation, or functional model supports the disease links. The heritability enrichment and colocalization are too high-level to generate actionable insight. This makes the manuscript appear opportunistic in scope, lacking a focused scientific question.
3. The use of an adjusted p-value threshold of 0.05 for MR results is relatively lenient, especially considering the large number of tests performed (>30,000 regulatory pairs). The authors must: Justify why genome-wide thresholds (e.g., 5e-8) were not applied. Provide sensitivity analyses with stricter thresholds (e.g., 1e-5, Bonferroni) to demonstrate robustness. Clarify how false positive rates were controlled in multi-trait colocalization.
4. The study aggregates eQTLs, mbQTLs, mQTLs, and GWAS hits, but the analytical path seems retrospective and unfocused-starting from large datasets and looking for associations, rather than addressing a precise biological hypothesis. As a result, the main message becomes diluted, and the manuscript lacks a compelling narrative arc or conceptual breakthrough.
5. Reporting tens of thousands of MR associations at a relaxed FDR threshold ($p < 0.05$) without replication or functional validation casts serious doubt on the true discovery rate. Without external validation or internal robustness checks, these results are likely overfitted to the input data and may not generalize.
6. Critical methodological steps, such as IV selection strategy, HEIDI filtering, and moloc posterior thresholding, are underexplained. Reproducibility and interpretability are hindered by this lack of transparency.
7. Given the large number of regulatory associations identified in this study, the authors are encouraged to compile the results into an interactive online database or web browser. This would not only increase the utility and visibility of the dataset but also allow other researchers to query specific gene-microbiome pairs and explore their relevance to traits or diseases. Such a resource would enhance the long-term impact of the work and facilitate validation, replication, and future functional studies.

Pre-decision referee cross-commenting

Reviewer #1

I think the author can solve these problems in their revisions or a new submit.

Actually, the authors used two-sample MR strategy to explore the relationships, so it's not a problem for the lack of control for individual-level confounding (Reviewer #2's concern)

Reviewer #2

This work presents a comprehensive analysis and is clearly written. I think it would be valuable to see a revised version that addresses the technical concerns outlined above.

Regarding the concern about individual-level confounding, I agree it should be acknowledged as a limitation. One way to partially address this could be to explore more stringent colocalization thresholds in a sensitivity analysis to assess the robustness of key associations. Additionally, incorporating MR methods robust to horizontal pleiotropy might help evaluate whether shared genetic effects are likely causal or confounded.

Reviewer #3

Thank you for the opportunity to contribute to this consultation. The manuscript represents a valuable and timely effort to integrate multi-tissue and multi-omics data to investigate gene-microbiome regulation and its relevance to human disease. While I remain concerned about the predominantly correlative nature of the findings and the lack of mechanistic insight, I believe these limitations can be transparently acknowledged and, at least partially, addressed through a major revision.

In particular, I would support a revision that includes:

More stringent statistical thresholds and sensitivity analyses to assess the robustness of key associations;

Greater methodological transparency, especially in the MR instrument selection, colocalization thresholds, and pleiotropy control;

Better contextualization of the findings within known disease biology, even if mechanistic models remain speculative at this stage.

Provided the authors are willing to address these concerns thoroughly, I would be supportive of giving them the opportunity to revise and resubmit.

Reviewer #1:

This paper systematically explores a multi-tissue interaction map between intestinal gene expression and gut microbiome composition. By integrating with GWAS datasets, it identified the gene-microbiome-disease regulatory loci, shedding light on the microbiome-dependent mechanism for disease loci indicative of the gut-organ axes. In addition, the analyses were comprehensive and in-depth. I feel it is worth reading by more researchers in the related fields. Several comments are as following:

Response: Thank you for the very positive comments on our manuscript and for appreciating the value of microbiome-dependent mechanisms in disease interpretation! We also greatly appreciate your very helpful suggestions below, which have helped us further improve our study. Please find our response detailed below.

1. As mentioned in line 377-379, "The three intestinal eQTL datasets are all from European ancestry, and the sample size of each is 368, 318, and 174, respectively". Are these sample sizes sufficient to ensure valid and reliable estimates in two-sample Mendelian randomization?

Response: Thank you for this important question! We fully agree that the sample size and statistical power of eQTLs can largely affect the reliability of two-sample MR analyses. To obtain the best power in integrative analyses with mbQTLs, we used GTEx intestinal eQTLs for MR identification, which represent the largest available sample sizes for intestinal tissues.

In fact, the GTEx consortium has conducted eQTL power analyses, showing that tissues with sample sizes of ~200 individuals yielded robust cis-eQTLs (GTEx Consortium, 2020), <https://gtexportal.org/home/methods>. Therefore, the eQTL sample sizes for the transverse colon ($N = 368$) and sigmoid colon ($N = 318$) should be sufficient, and ileum ($N = 174$) should be nearly sufficient. Consistently, the number of significant MR pairs identified (in the revised manuscript) is 6,088 and 5,810 for transverse colon and sigmoid colon, respectively, and 2,398 for ileum. While the smaller ileum sample size ($N = 174$) resulted in relatively fewer significant pairs, transverse colon ($N = 368$) and sigmoid colon ($N = 318$) yielded comparable numbers, suggesting that sample sizes above ~300 may approach saturation for MR identification.

To make it transparent to the readers, we have discussed this potential limitation regarding eQTL sample sizes in the revised manuscript, which reads as:

"Leveraging the MR approach, we identified 6,088 gene-to-microbiome regulatory pairs in transverse colon, 5,810 in sigmoid colon, and 2,398 in ileum, respectively, substantially expanding the repertoire of regulatory pairs. The relatively smaller number of regulatory pairs identified in ileum was likely due to the smaller eQTL sample size ($N = 174$) compared to transverse colon ($N = 368$) and sigmoid colon ($N = 318$)".

2. Is the use of multiple-trait colocalization alone sufficient to infer that the microbiome mediates the effect of gene expression on disease? Additional evidence may be

Response: Thank you for catching these typos! We have now used a consistent format for figure citations throughout the manuscript (Fig. 2A, Fig. 2B, Fig. 3G, etc).

4. The pronounced regulatory cluster on chromosome 12 is interesting. Could the authors explain its potential biological significance?

Response: Thank you for pointing out the pronounced regulatory cluster on chromosome 12! Indeed, GWAS studies have revealed that chromosome 12 possesses various genetic risk loci that are associated with intestinal diseases, particularly Crohn's disease (CD) and ulcerative colitis (UC) (Lesage *et al*, 2000; Lin *et al*, 2020; Herrick & Tansey, 2021). To further understand the potential biological significance of this regulatory cluster, we carried out functional enrichment analyses for the genes with significant MR regulatory pairs on chromosome 12. Intriguingly, we found pathway enrichments relevant to the intestinal and microbial pathways, such as regulation of calcium-mediated signaling (Loh *et al*, 2024), sensory perception of taste (Depoortere, 2014), and serine family amino acid catabolic process (Mardinoglu *et al*, 2015; Li *et al*, 2024a) (**Revised Figure EV1E**, also pasted below). These results suggest that the pronounced regulatory cluster on chromosome 12 may underlie the interaction between intestinal pathways and microbiome. We have added this result to the revised manuscript.

Revised Figure EV1E. The functional annotation of the genes from the regulatory MR pairs on chromosome 12 in three tissues.

5. Please revise "Collection of eQTL, mQTL, and mQTL datasets" to "Collection of eQTL, mQTL, and mbQTL datasets".

Response: Thank you for pointing it out! We have revised it accordingly.

We would like to thank you again for these helpful comments and suggestions!

Reviewer #2:

Overall Summary: Wang et al. investigate how host genetic regulation of intestinal gene expression influences gut microbiome composition and potentially contributes to complex human diseases. Using two-sample Mendelian Randomization and colocalization analyses, the authors link eQTLs (from GTEx gut tissues) with microbiome QTLs (from MiBioGen) and 89 GWAS datasets. They construct tissue-specific gene-to-microbiome regulatory maps and identify immune- and metabolism-related pathways mediate gene-to-microbiome regulation across intestinal tissues. They found that gene-to-microbiome regulatory SNPs significantly contribute to the heritability of multiple complex diseases, particularly for respiratory and neuropsychiatric disorders, and identify microbial taxa such as Bifidobacterium and Ruminococcaceae that potentially mediate these effects.

This is a well written manuscript and addresses an important question in the field. However, there are a few concerns that need to be addressed:

Response: We thank you for appreciating the value of our study! We also sincerely appreciate your constructive comments and suggestions, based on which we carried out more stringent MR identification and performed experimental validations in mice to strengthen our manuscript. Please find our detailed response below.

Major critiques:

1. The study uses two entirely separate datasets (GTEx for eQTLs and MiBioGen for mbQTLs), which means gene expression and microbiome are never observed in the same individual. This prevents control for individual-level confounding, environmental exposures, such as medication or diet, which could independently influence both gene expression and the microbiome. Thus, the biological variation across tissues/cohorts may lead to misleading gene-microbiome regulatory associations.

Response: Thank you for bringing up this important technical detail! We agree with you that accounting for individual-level confounders would be difficult given that we utilized two entirely separate datasets for MR analysis. Therefore, we chose the two-sample MR strategy for identifying potential gene-to-microbiome regulatory pairs, which requires the eQTLs and mbQTLs to be derived from two separate study populations to mitigate potential confounding factors (Davies *et al*, 2018).

We also appreciate your suggestions on this point in the cross-commenting section. Following these helpful suggestions, we optimized our procedure for identifying gene-to-microbiome regulatory pairs in multiple aspects:

- 1) We applied a more stringent threshold (P value $< 5 \times 10^{-8}$) on eQTLs for selecting instrumental variables (IVs);
- 2) We excluded the MR pairs inferred from single IVs and only retained those obtained from multiple IVs to increase the reliability;

- 3) We utilized the MR-Egger and HEIDI tests to exclude those pairs showing potential pleiotropic effects;
- 4) We conducted the leave-one-out analysis to assess whether any causal effect estimates are driven by one SNP independently.

In addition, we have also discussed the potential confounding effects in our analysis in the revised manuscript, which reads as:

“Moreover, since the eQTLs and mbQTLs were collected from two separate cohorts, we utilized a two-sample MR framework to mitigate potential individual-level confounders. Future studies focusing on a single cohort with a comprehensive curation of phenotypes can help disentangle gene-to-microbiome regulation from potential confounders”.

2. How was the threshold of $PP4 > 0.5$ for colocalization determined? This value is relatively permissive and may not provide strong evidence for shared causal variants. It would be helpful to know whether any sensitivity analyses were conducted to assess the robustness of this threshold. Adopting a more stringent criterion or validating the findings in independent datasets would strengthen the evidence for shared genetic regulation.

Response: We appreciate these important technical suggestions! Following your advice, we further conducted sensitivity tests (as detailed in our response #1 above), as well as adopting a more stringent $PP4$ criterion for colocalization analyses and utilizing an independent eQTL dataset for external validation:

1. We initially chose $PP4 > 0.5$ as the threshold because it has been frequently adopted in various authoritative genetic studies (Roychowdhury *et al*, 2023; Kim-Hellmuth *et al*, 2020; GTEx Consortium, 2020; Arvanitis *et al*, 2022; Gay *et al*, 2020). Nevertheless, we agree with you that a more stringent $PP4$ cutoff will increase the robustness of our conclusions; therefore, we further defined “tier 1” colocalization loci using $PP4 > 0.9$, in addition to the original $PP4 > 0.5$ loci (defined as tier 2) in the revised manuscript. Moreover, as mentioned above, we performed sensitivity tests to exclude any possible pleiotropic effects for the regulatory loci. These will collectively increase the reliability and transparency of the gene-to-microbiome regulatory loci that we identified.
2. To further strengthen our results, we collected transverse colon eQTLs from the CEDAR database as an independent validation dataset (Momozawa *et al*, 2018). Of the 285 eQTL-mbQTL colocalized loci we identified in transverse colon, only 37 can be evaluated since other loci do not have significant eQTLs in the CEDAR data, likely due to the population difference and the relatively limited statistical power. However, regardless of technical limitations, 51% (19 out of 37) of the tested loci can be independently validated.

We have added these analyses and new results into the revised manuscript. Thank you for these suggestions that helped strengthen the rigor of our analyses.

3. The paper focuses on statistical significance, but rarely discusses effect sizes or

confidence intervals which impedes the assessment of biological magnitude of gene-to-microbiome effects.

Response: We appreciate your valuable comments regarding the assessment of the biological magnitude, in addition to statistical significance, of gene-to-microbiome regulation. Following your suggestions, we performed multiple analyses during revision, including: 1) the consistency of the effect sizes across three intestinal tissues (**Revised Figure 3D, Revised Figure EV3B**); 2) we compared the effect sizes of gene-to-microbiome regulatory pairs with regard to different categories (**Revised Figure 1D, Revised Figure EV1C, Revised Figure EV2B and Revised Figure EV3C**). Specifically:

- A. We showed the effect consistency (from 99.3% to 100%) of the regulatory directionalities in the significant gene-to-microbiome MR pairs across three tissues (**Revised Figure 3D**, also pasted below), indicating the robustness of gene expression's effect on microbiome composition.

Revised Figure 3D. Pairwise comparisons of the effect sizes of the gene-to-microbiome regulatory pairs across three tissues. Pearson correlation coefficients and the corresponding *P* values were calculated. The directionality consistency for each pairwise comparison is shown at the top of each panel.

- B. We compared the overall effect sizes of gene-to-microbiome regulation across different genera, and showed the top 20 genera being most strongly affected by gene expression in the **Revised Figure 1D** (also pasted below, for transverse colon) and **Figure EV1D** (for the other two tissues). Among the top 20 genera, we found that 70% (14 out of 20) genera were shared between the three tissues, demonstrating that the magnitude of gene-to-microbiome are also consistent across tissues.

Revised Figure 1D. The effect sizes of the gene-to-microbiome regulation of the top 20 genera in the transverse colon. The y-axis indicates the absolute effect size, and the x-axis indicates the top 20 genera with the strongest overall regulatory magnitude.

C. We further compared the regulatory magnitude of genes with broad effects on more than 10 different genera (“broadly regulating” group) versus those with more specific regulatory effects on fewer genera (“specifically regulating” group). We found that the specifically regulating group exhibited higher regulatory effects compared to the regulatory pairs in the broadly regulating group (**Revised Figure EV2B**, also pasted below).

Figure EV2B. Comparison of the overall effect sizes of gene-to-microbiome pairs between broadly regulating groups and specifically regulating groups in the three tissues.

D. Lastly, we evaluated the overall magnitude of the gene-to-microbiome regulatory

pairs in different tissue-sharing and tissue-specific groups (**Revised Figure EV3A**, also pasted below). We found that transverse-colon-specific regulatory groups showed the highest regulatory effect, while the group of MR pairs shared across three tissues exhibited the lowest regulatory effect.

Revised Figure EV3A. Comparison of the effect sizes in the tissue-sharing and tissue-specific groups, with x-axis illustrating the MR regulatory types, and y-axis showing the absolute value of the effect size.

We have added these results and figures to the revised manuscript. Thank you again for this constructive suggestion that ensured both the statistical significance and regulatory magnitude were investigated in this study.

4. The functional interpretation of genera (e.g., SCFA production by *Allisonella* or *Ruminococcaceae* UCG003) relies on taxonomic annotations rather than direct functional or metabolomic data. Such genus-level inferences are speculative, as microbial functions are often strain-specific and context-dependent, requiring validation through metagenomics or metabolomics.

Response: We appreciate your suggestion regarding the functional validation of our findings. We agree that direct functional support is needed beyond the taxonomic annotations. Therefore, we followed your suggestion and carried out experiments to validate the regulatory roles of different microbial strains on SCFA production.

In the revised manuscript, we found that *Allisonella*, *Blautia*, and *Dorea* genera were among the top genera that possessed the largest number of significant regulatory pairs, with previous studies suggesting that they were all associated with short-chain fatty acid (SCFA) production (Ecklu-Mensah *et al*, 2023; Holmberg *et al*, 2024; Nishiwaki *et al*, 2022). To directly support the causal roles of these genera in SCFA production, we carried out microbial colonization experiments in mice and examined the resulting change in SCFA levels. For each of these three genus, we selected one representative strain that is commercially available. Microbial colonization experiments in mice showed that all three treatment groups exhibited significantly upregulated SCFA levels

compared to the control groups (revised **Figure EV4C**, also pasted below). Therefore, our results validate the convergent function of different microbiota on the SCFA production.

We have added these new results into the revised manuscript. Thank you for this helpful suggestion that improved the reliability of these results.

Revised Figure EV4C. Experimental validations for the regulatory effects of different microbiota on SCFA production.

5. The tissue-specificity analysis compares tissue-specific vs. shared gene-to-microbiome regulatory pairs across intestinal sites, but does not fully control for the differing number of eQTLs, sample sizes, or microbiome MR pairs per tissue. As a result, the apparent tissue specificity could partly reflect differences in detection power rather than true biological differences in regulatory architecture.

Response: We agree that the different sample sizes and detection power of GTEx eQTLs between different tissues can partly contribute to the observed tissue specificity of the identified gene-to-microbiome regulation.

Since the effect size is relatively less affected by detection power compared to P values (Hou *et al*, 2023; Xiong *et al*, 2021), we evaluated the potential tissue specificity by examining the effect size consistency between tissues for the tissue-specific regulatory pairs. While these pairs were statistically significant in only one tissue, their effect sizes and directionalities remained consistent with other tissues ($R^2 = 0.78$, revised **Figure EV3C**, also pasted below). Notably, the regression coefficient was 0.81, suggesting that the effect sizes in the non-significant tissues were relatively weaker compared to the tissue where significance was detected. This result indicated that these tissue-specific regulatory pairs maintain a certain degree of biological consistency in other tissues, but with a lower effect magnitude.

Collectively, as the reviewer mentioned, the tissue specificity of gene-to-microbiome partly reflected the difference in detection power, but could also be partly due to the lower effect magnitude and hence the insignificant P values in other tissues. To make it

transparent to the audience, we have added these new results to the revised main text, and further discussed the detection power limitation and the tissue-specific regulation in the revised Discussion section, which reads as:

For the detection power limitation, we wrote: “*The relatively smaller number of regulatory pairs identified in ileum was likely due to the smaller eQTL sample size (N = 174) compared to transverse colon (N = 368) and sigmoid colon (N = 318)*”.

For the tissue-specific regulatory pairs, we wrote: “*Of note, given that the tissue-specific regulatory pairs maintained an overall consistent but reduced effect size in the non-significant tissues, the observed tissue specificity was likely a combined effect of statistical power and biological difference*”.

Thank you for improving the rigor of our analyses and conclusions regarding the tissue specificity of gene-to-microbiome regulation.

C

Comparison of effect sizes for tissue-specific regulatory pairs in three tissues

Revised Figure EV3C. Comparisons of the effect sizes for the tissue-specific regulatory pairs between the significant tissue versus other tissues. Regression coefficient, Pearson correlation coefficients, and the corresponding P values were calculated.

6. What background genes were used for pathway enrichment analysis?

Response: Thank you for this technical question. We used the background gene lists that matched the biological contexts for GO analyses to avoid potential tissue-specific or statistical significance bias. For instance:

1. For the functional enrichment analysis of the genes that regulate the microbiome in each tissue, such as the results shown in **Figure 2D-E** and **Figure EV2F**, we used all the genes that were tested in the MR analysis, regardless of significance, in the corresponding tissue as the background gene list.
2. For the functional enrichment analysis of the genes within different tissue-sharing groups, such as results shown in **Figure 3F** and **Figure EV3D**, we used all the genes that were detected as significant MR pairs in all the three tissues as the background gene list to reflect the functional enrichment between different tissue-sharing groups.

We agree with you that the selection of background genes is important for the reliability of GO analysis. Therefore, to ensure the transparency of our analysis, we have added these technical details to the **Methods** section, which reads as:

“For the GO enrichment analysis of the genes involved in the “broadly-regulating” and “specifically-regulating” across three tissues, all genes tested in the MR analysis for each respective tissue were used as the background set, regardless of significance. For the GO and DO enrichments of the genes in the different tissue-sharing and tissue-specific groups, the background gene list consisted of all genes identified as significant MR pairs across the three tissues”.

Minor critiques:

- In Fig 4C heatmap, abbreviations for traits (e.g., FVC, PEF) are used without immediately defining them in the figure or caption.

Response: We have defined these abbreviations by their full names in the revised caption of **Figure 4C**. Thank you for pointing it out.

- In Fig 4D,E, the abbreviation "PP abc" is used without defining it in the main text or caption.

Response: We have defined “PP abc” in both the main text and the figure caption.

We would like to thank you again for all the very constructive comments that helped greatly improve the manuscript!

Reviewer #3:

Summary

The authors present a computational study aiming to characterize the genetic regulation of the gut microbiome via intestinal gene expression across three tissues (terminal ileum, transverse colon, and sigmoid colon). Leveraging Mendelian randomization (MR), colocalization, and methylation QTL data, they identify over 30,000 gene-microbiome regulatory pairs, some of which are further linked to disease heritability via GWAS loci and colocalization. The study proposes that tissue-specific and shared gene-microbiome interactions may contribute to inter-organ disease mechanisms such as the

gut-lung and gut-brain axes.

Key conclusions include:

Thousands of host genes regulate gut microbiome genera in a tissue-specific or tissue-shared manner.

Certain genes (e.g., ATG13, A2M, RBM6) are implicated as microbiome-mediated contributors to disease loci via colocalization and trait enrichment.

Microbiome regulation may occur via gene expression and epigenetic modifications, and may influence systemic diseases through microbiome shifts.

The authors use GTEx-derived eQTLs from intestinal tissues, MiBioGen microbiome QTLs, methylation QTLs, and GWAS summary statistics. The main methodologies are two-sample MR, coloc, and multi-trait colocalization (moloc).

General Remarks

This is an ambitious manuscript that integrates several data modalities to investigate the genetic regulation of the gut microbiome and its relevance to human disease. The tissue-specific aspect and the inclusion of methylation data are valuable extensions to existing host-microbiome QTL analyses.

However the authors provide an extensive integrative analysis across multiple tissues and data types, the work lacks clear mechanistic insight and suffers from weak statistical rigor. The main findings are associative, not causal, and several methodological decisions (e.g., p-value thresholds, lack of validation) significantly weaken the robustness and interpretability of the results. Particularly, the gene-microbiome-disease axes remain mostly speculative without functional support or deeper mechanistic insight. Additionally, several methodological choices require further justification or clarity. If the authors can address the concerns below-especially regarding statistical stringency, interpretation of disease relevance, and validation of key findings-this work would be significantly strengthened.

Response: We appreciate your positive comments and your very constructive remarks. Following your suggestions, we substantially revised our manuscript by improving statistical rigor, providing in-depth biological interpretation, performing various functional validations with mouse experiments, and constructing an interactive online database. Please find our detailed response below.

Major Concerns

1. While the manuscript reports thousands of associations between gene expression and microbiome features, it remains largely correlative in nature. There is no mechanistic model proposed, no clear causal chain established, and no experimentally testable hypothesis generated. The work functions more as a catalog than a mechanistic study.

Response: We thank you for this insightful comment. Based on your suggestions, we

have substantially revised our manuscript by adding in-depth mechanistic interpretations and carrying out multiple mouse experiments, thereby establishing a clearer causal chain across gene expression, microbiome, and human diseases. Specifically:

1. Regarding a clearer causal relationship, we performed a bi-directional MR analysis between intestinal gene expression and microbiome to distinguish the potential regulatory direction from gene expression to microbiome versus the reverse (please see the **revised Figure 1B** and **Figure EV1F**).
2. We also extended beyond the simple correlation analysis and provided mechanistic interpretations of diseases with experimental support. Specifically, we delved into two specific disease loci to build up causal models by which gene-to-microbiome regulation contributes to depression and allergic rhinitis.
 - a. In the first disease locus, we established the causal role that *Allisonella* suppresses depression through regulating the concentration of a metabolite bile acid (BA) using mouse experiments. The new results were added as revised **Figure 4E-G**, and more details can be found in our response to your comment #2 below.
 - b. In the second disease locus, we experimentally demonstrated the regulation of *Parasutterella* on short-chain fatty acid (SCFA) production using mouse colonization assays, thereby providing a mechanistic link to the regulatory roles of *Parasutterella* on allergic rhinitis. The new results were added as revised **Figure EV4G-H**, and more details can be found in our response to your comment #2 below as well.

With these newly added mechanistic interpretations and experimental validations, we have now provided more mechanistic insights into the relationship across gene expression, microbiome, and disease. On the other hand, we also agree that our work also functions as a catalog, and therefore have compiled the established regulatory maps into an interactive web browser, as you kindly suggested in comment #7 (<https://xiongqslab.github.io/microbiomeMR/>).

2. The manuscript claims relevance to disease via GWAS integration, but the connections are shallow: No disease mechanism is unpacked in depth. No follow-up analysis, validation, or functional model supports the disease links. The heritability enrichment and colocalization are too high-level to generate actionable insight. This makes the manuscript appear opportunistic in scope, lacking a focused scientific question.

Response: Thank you for the suggestion to dig more into disease mechanisms! As you suggested, we now carried out more follow-up disease analysis, unpacked two specific disease loci in depth, carried out experimental validations, and proposed functional models.

We selected two representative disease loci and carried out mouse experiments to dive into the potential mechanisms.

1) In the first locus, we observed a multi-trait colocalization signature across *ZNF584* gene, *Allisonella* genus, and depression GWAS (**Figure 4D**), which was further supported by our two-step MR analysis. Given the previously established role of bile acid (BA) in depression (Klein & Kheirbek, 2024; Li *et al*, 2024b) and our predicted relationship between *Allisonella* and depression, we next explored the potential causal chain from *Allisonella* composition to BA production, and eventually to depression. We first treated mice with chronic restraint stress (CRS), a commonly used strategy for constructing depression mouse models, and observed significant reductions in both *Allisonella* abundance (P value = 0.028) and BA level (P value = 0.014) (please see the revised **Figure 4E**, also pasted below), consistent with the known suppressive effect of BA on depression. We then further explored the relationship between *Allisonella* abundance and BA level by mouse oral gavage of *Allisonella* with different concentrations (OD600 = 0.4 and 0.6), and found that *Allisonella*-treated mice exhibited dosage-sensitive increase of BA levels compared to the control group (please see the revised **Figure 4F**, also pasted below). Collectively, building up on the established role of BA in suppressing depression, our analyses and experiments further filled a regulatory gap where CRS treatment contributes to depression via *Allisonella*-mediated regulation on BA production (please see the revised **Figure 4G**, also pasted below).

Revised Figure 4E. *Allisonella* species abundance (left) and bile acid (BA) level (right) in the CRS-treated and control mice.

Revised Figure 4F-G. Comparisons of the BA concentration between control mice and mice treated with different concentrations of *Allisonella* (F). Schematic illustrating the regulatory axis across CRS treatment, *Allisonella* abundance, BA level, and depression (G).

- 2) In the second case, we focused on a multi-trait colocalization locus across *ABHD12* gene, *Parasutterella*, and allergic rhinitis. Intriguingly, previous studies have indicated the association between *Parasutterella* abundance and allergic rhinitis (Yang *et al*, 2022), as well as the regulatory role of short-chain fatty acid (SCFA) in allergic rhinitis (Zhou *et al*, 2021; Liu *et al*, 2023). However, the exact relationships across *Parasutterella*, SCFA abundance, and allergic rhinitis remained unclear given the regulatory gap between *Parasutterella* and SCFA. To address this, we performed microbiome-colonization experiments in mice using two different concentrations of *Parasutterella* treatments. We observed significantly increased levels of acetate, a major type of SCFA, in the *Parasutterella*-treated mice, with the higher treatment concentration inducing a stronger SCFA elevation ($P = 1 \times 10^{-3}$ for OD0.4 and 4.9×10^{-5} for OD0.6, please see the **revised Figure EV4G**, also pasted below). Therefore, our results established the regulation of *Parasutterella* on SCFA, providing a mechanistic link to their regulatory roles in allergic rhinitis (**revised Figure EV4H**, also pasted below).

Revised Figure EV4G-H. Comparisons of the SCFA (acetate) concentration between control mice and mice treated with different concentrations of *Parasutterella* (**G**). Schematic illustrating the regulatory axis across *Parasutterella* abundance, SCFA level, and allergic rhinitis (**H**).

We have added the results to the section “Gene-to-microbiome regulation contributes to disease heritability” in the revised manuscript, and illustrated the potential working models using schematic plots (revised **Figure 4G** and **Figure EV4H** as shown above). Thank you again for this helpful suggestion!

3. The use of an adjusted p-value threshold of 0.05 for MR results is relatively lenient, especially considering the large number of tests performed (>30,000 regulatory pairs). The authors must: Justify why genome-wide thresholds (e.g., $5e-8$) were not applied. Provide sensitivity analyses with stricter thresholds (e.g., $1e-5$, Bonferroni) to demonstrate robustness. Clarify how false positive rates were controlled in multi-trait colocalization.

Response: We thank you for these important technical comments and suggestions! Our improved analyses and detailed response are as follow:

1. Regarding the threshold for defining significant MR pairs, we utilized a study-wise P value correction to account for the number of multiple tests performed. While genome-wide P value threshold (5×10^{-8}) was a commonly-used standard in genetics association studies, as you pointed out, it is predominantly adopted in the GWAS analysis, where typically millions ($>1 \times 10^6$) of statistical tests are performed. In our study, we only performed tests for those having significant eQTLs as IVs, and therefore utilized a multi-testing correction to account for our statistical burden rather than using the genome-wide threshold.

2. Following your suggestion, we performed sensitivity tests to exclude any possible pleiotropic effects and to improve the robustness of regulatory pairs. For the filtering of MR pairs using sensitivity tests, a more stringent threshold will lead to the retaining of more potential false regulatory pairs (*i.e.* a more significant P value indicates higher probability of being a problematic locus). Therefore, we utilized relative lenient P value thresholds (unadjusted P value < 0.05 for MR-Egger test) to make sure that MR pairs with even nominal significance against sensitivity test were excluded. In addition, we further improve the stringency of MR identification in other aspects:
 - a. We applied a more stringent threshold (P value $< 5 \times 10^{-8}$) on intestinal eQTLs for selecting instrumental variables (IVs).
 - b. We excluded the MR pairs inferred from single IVs and only retained those obtained from multiple IVs.
 - c. We performed a leave-one-out analysis to remove MR pairs that might be driven by only one SNP independently (Please see the revised **Methods** and **revised Figure 1B**).

We believe these together largely increased the reliability of the identified gene-to-microbiome regulatory pairs.

3. We also appreciate your question regarding the control of false positives in multi-trait colocalization analysis. However, this has been a technical difficulty in the field given the complexity of different genetic configurations, not only for multi-trait colocalization but even for colocalization between only two traits (Wang *et al*, 2022; Wellcome Trust Case Control Consortium, 2007; Wacholder *et al*, 2004). In fact, studies in the field barely considered multiple hypothesis testing for colocalization analysis: “*Although the threshold for the posterior probability can be modified to account for multiple hypothesis testing by adjusting the prior probability and other factors in the calculation of false positive report probability (FPRP), explicit recommendations to control the false positive rate when testing multiple genes and/or tissues at a locus for these Bayesian colocalization methods have not been detailed*”. – quoting Fan Wang *et al.*, *AJHG*, 2022.

However, to make it transparent to the readers, we discussed this potential limitation in our revised **Discussion** section and quoted the relevant papers accordingly. It reads as: “*While we accounted for multiple testing by P value corrections for the majority of our analyses, it was technically difficult for multi-trait colocalization analysis*”.

4. The study aggregates eQTLs, mbQTLs, mQTLs, and GWAS hits, but the analytical path seems retrospective and unfocused-starting from large datasets and looking for associations, rather than addressing a precise biological hypothesis. As a result, the main message becomes diluted, and the manuscript lacks a compelling narrative arc or conceptual breakthrough.

Response: We thank you again for this valuable comment! As described in detail in our

response to your comments #1 and #2 above, we have moved beyond simple associations to focus on the potential mechanisms of two illustrative disease loci, including the *Allisonella*-BA-depression and *Parasutterella*-SCFA-allergic rhinitis regulatory axes. Our revised manuscript now more clearly illustrates how we may start from large datasets and then investigate specific disease mechanisms. We have added these two disease loci as representative examples in the **revised Abstract**.

5. Reporting tens of thousands of MR associations at a relaxed FDR threshold ($p < 0.05$) without replication or functional validation casts serious doubt on the true discovery rate. Without external validation or internal robustness checks, these results are likely overfitted to the input data and may not generalize.

Response: We agree with you that a stringent analysis procedure and an external validation are both important to ensure the reliability of these MR associations. Based on your suggestions, we carefully revised our analyses as follows.

1. As partly mentioned in our response to your comment #3 above, we optimized the procedure for identifying gene-to-microbiome regulatory pairs in multiple aspects:
 - a. We applied a more stringent threshold (P value $< 5 \times 10^{-8}$) on eQTLs for selecting instrumental variables (IVs).
 - b. We excluded the MR pairs inferred from single IVs and only retained those obtained from multiple IVs to increase the reliability.
 - c. We performed sensitivity tests to exclude those pairs showing potential pleiotropic effects.
 - d. We conducted the leave-one-out analysis to assess whether any causal effect estimates are driven by one SNP independently.

This stringent procedure reduced the number of identified regulatory pairs from the originally reported range of 6,840–13,419 to 2,398–6,088 across three tissues, but with a markedly improved reliability.

2. Regarding the external validation, we further collected transverse colon eQTL dataset from the CEDAR database (Momozawa *et al*, 2018) to evaluate the gene-to-microbiome regulatory pairs we identified in the transverse colon. While only 767 gene-microbiome regulatory pairs can be tested due to the availability of significant eQTLs as instrumental variables in the CEDAR data, 295 (38.5%) regulatory pairs can be validated in the replication stage. Notably, in addition to the significant pairs replicated, the regulatory pairs identified using the two different datasets demonstrated strong effect size consistency ($R^2 = 0.68$, P value $< 2.2 \times 10^{-16}$, please see the revised **Figure EV1B**, also pasted below). Therefore, both the direct overlap and effect size consistency analyses suggested the robustness of the regulatory pairs identified, especially considering the different statistical power and different genetic background between the GTEx and CEDAR cohorts.

Thank you for this important question that has helped ensure the rigor of our regulatory pair identification and analysis.

Revised Figure EV1B. Effect size consistency the MR regulatory pairs identified based on GTEx (discovery) versus CEDAR (replication) datasets.

6. Critical methodological steps, such as IV selection strategy, HEIDI filtering, and moloc posterior thresholding, are underexplained. Reproducibility and interpretability are hindered by this lack of transparency.

Response: As you kindly suggested, we added the detailed descriptions of these analyses in the **Methods** section. Specifically:

- 1) For instrumental variable selections, we wrote: “For instrumental variables (IVs), we selected significant eQTLs defined by GTEx that were after two rounds of multiple testing corrections, and further applied a P value threshold of 5×10^{-8} to increase the reliability of the MR analysis. Subsequent LD clumping with a strict threshold ($r^2 < 0.01$ within 100 kb windows) was conducted to pick out independent instruments using 1000 Genomes Project phase 3 as the reference panel”.
- 2) For HEIDI test filtering, we wrote: “We conducted the heterogeneity in dependent instruments (HEIDI) test to determine whether the causal relationship from gene expression to microbial abundance was driven by a shared genetic variant or a linkage model. MR results with a significant P_{HEIDI} value under 0.01 were filtered out for further analysis”.
- 3) For Moloc analysis, we wrote: “Moloc computed the posterior probabilities for 15 possible configurations of the associations. A threshold of 0.1 for the posterior probability colocalization (PP abc) across eQTL, mbQTL, and GWAS was utilized”.
- 4) For additional MR filtering, we wrote: “We excluded the MR pairs inferred from single IVs and only retained those obtained from multiple IVs. In addition, we utilized the MR-Egger test to exclude those pairs showing potential pleiotropic

effects (P value < 0.05). We further conducted the leave-one-out analysis to assess whether any causal effect estimates are driven by one SNP independently”.

In addition, we have deposited all the relevant codes for these analyses in the github (<https://github.com/xiongxslab/microbiomeMR>). Thank you for this suggestion that helped increase the transparency of these analyses!

7. Given the large number of regulatory associations identified in this study, the authors are encouraged to compile the results into an interactive online database or web browser. This would not only increase the utility and visibility of the dataset but also allow other researchers to query specific gene-microbiome pairs and explore their relevance to traits or diseases. Such a resource would enhance the long-term impact of the work and facilitate validation, replication, and future functional studies.

Response: Thank you for this helpful suggestion! We have assembled an interactive online database (<https://xiongxslab.github.io/microbiomeMR/>) that allows users to query, download, and explore gene-to-microbiome regulatory pairs along with associated traits and diseases. As you suggested, we believe this database can increase the accessibility and long-term utility of our results, thereby facilitating validation, replication, and future functional studies. We have added the web link to the abstract.

We would like to thank you again for the above insightful comments and suggestions, which have truly helped increase the rigor and biological significance of this study.

Pre-decision referee cross-commenting

Reviewer #1

I think the author can solve these problems in their revisions or a new submit.

Actually, the authors used two-sample MR strategy to explore the relationships, so it's not a problem for the lack of control for individual-level confounding (Reviewer #2's concern)

Response: Thank you for the positive comments on our study! As detailed above, we have carefully addressed the very helpful suggestions from both you and other reviewers in the revised manuscript.

Reviewer #2

This work presents a comprehensive analysis and is clearly written. I think it would be valuable to see a revised version that addresses the technical concerns outlined above.

Regarding the concern about individual-level confounding, I agree it should be acknowledged as a limitation. One way to partially address this could be to explore more stringent colocalization thresholds in a sensitivity analysis to assess the

robustness of key associations. Additionally, incorporating MR methods robust to horizontal pleiotropy might help evaluate whether shared genetic effects are likely causal or confounded.

Response: Thank you for appreciating the value of our study! We have carefully addressed both the biological and technical concerns from you and other reviewers.

We also appreciate your additional comments on individual-level confounding and for your suggested solutions. Following your suggestions, we have: **i)** discussed the potential individual-level confounding issue in our revised **Discussion** section; **ii)** applied more stringent thresholds for both MR and colocalization analyses with sensitivity tests; **iii)** utilized MR-Egger and HEIDI tests to control for those pairs with potential pleiotropic effects. Please see our response to your comment #1 for more details.

Reviewer #3

Thank you for the opportunity to contribute to this consultation. The manuscript represents a valuable and timely effort to integrate multi-tissue and multi-omics data to investigate gene-microbiome regulation and its relevance to human disease. While I remain concerned about the predominantly correlative nature of the findings and the lack of mechanistic insight, I believe these limitations can be transparently acknowledged and, at least partially, addressed through a major revision.

Response: Thank you for appreciating the value and effort of our study! As detailed in our response above, we have increased the stringency of our analysis and performed multiple experimental validations to strengthen causal relationships and to provide mechanistic insights for gene-microbiome regulation and disease relevance.

We also thank you for providing the very specific guidance listed below for addressing the key concerns. Specifically:

In particular, I would support a revision that includes:

More stringent statistical thresholds and sensitivity analyses to assess the robustness of key associations;

We have utilized more stringent statistical thresholds, and carried out sensitivity analyses to improve the reliability of the regulatory pairs we identified. Please see our response to your comments #3 and #5 for details.

Greater methodological transparency, especially in the MR instrument selection, colocalization thresholds, and pleiotropy control;

We have added detailed descriptions of these analyses in the **Methods** section to increase methodological transparency, including for the selection of instrumental variables for MR. Please see our response to your comment #6 for details.

Better contextualization of the findings within known disease biology, even if mechanistic models remain speculative at this stage.

We have now contextualized multiple gene-microbiome regulatory loci into known disease biology, and further focused on two disease axes that involved depression and allergic rhinitis by carrying out mouse experiments. Please see our response to your comment #1 and #2 for details.

Provided the authors are willing to address these concerns thoroughly, I would be supportive of giving them the opportunity to revise and resubmit.

We would like to sincerely thank all the very helpful and constructive comments from the three reviewers! We believe the revised manuscript has been substantially improved.

Reference:

Arvanitis M, Tayeb K, Strober BJ & Battle A (2022) Redefining tissue specificity of genetic regulation of gene expression in the presence of allelic heterogeneity. *Am J Hum Genet* 109: 223–239

Davies NM, Holmes MV & Davey Smith G (2018) Reading Mendelian randomisation studies: a guide, glossary, and checklist for clinicians. *BMJ* 362: k601

Depoortere I (2014) Taste receptors of the gut: emerging roles in health and disease. *Gut* 63: 179–190

Ecklu-Mensah G, Choo-Kang C, Maseng MG, Donato S, Bovet P, Viswanathan B, Bedu-Addo K, Plange-Rhule J, Oti Boateng P, Forrester TE, *et al* (2023) Gut microbiota and fecal short chain fatty acids differ with adiposity and country of origin: the METS-microbiome study. *Nat Commun* 14: 5160

Gay NR, Gloudemans M, Antonio ML, Abell NS, Balliu B, Park Y, Martin AR, Musharoff S, Rao AS, Aguet F, *et al* (2020) Impact of admixture and ancestry on eQTL analysis and GWAS colocalization in GTEx. *Genome Biol* 21: 233

GTEx Consortium (2020) The GTEx Consortium atlas of genetic regulatory effects across human tissues. *Science* 369: 1318–1330

Herrick MK & Tansey MG (2021) Is LRRK2 the missing link between inflammatory bowel disease and Parkinson's disease? *NPJ Parkinsons Dis* 7: 26

Holmberg SM, Feeney RH, Prasoodanan P K V, Puértolas-Balint F, Singh DK, Wongkuna S, Zandbergen L, Hauner H, Brandl B, Nieminen AI, *et al* (2024) The gut commensal *Blautia* maintains colonic mucus function under low-fiber consumption through secretion of short-chain fatty acids. *Nat Commun* 15: 3502

Hou L, Xiong X, Park Y, Boix C, James B, Sun N, He L, Patel A, Zhang Z, Molinie B, *et*

- al* (2023) Multitissue H3K27ac profiling of GTEx samples links epigenomic variation to disease. *Nat Genet* 55: 1665–1676
- Kim-Hellmuth S, Aguet F, Oliva M, Muñoz-Aguirre M, Kasela S, Wucher V, Castel SE, Hamel AR, Viñuela A, Roberts AL, *et al* (2020) Cell type-specific genetic regulation of gene expression across human tissues. *Science* 369
- Klein AS & Kheirbek MA (2024) From bile acids to melancholia. *Neuron* 112: 1725–1727
- Lesage S, Zouali H, Colombel JF, Belaiche J, Cézard JP, Tysk C, Almer S, Gassull M, Binder V, Chamaillard M, *et al* (2000) Genetic analyses of chromosome 12 loci in Crohn's disease. *Gut* 47: 787–791
- Lin L, Zhou G, Chen P, Wang Y, Han J, Chen M, He Y & Zhang S (2020) Which long noncoding RNAs and circular RNAs contribute to inflammatory bowel disease? *Cell Death Dis* 11: 456
- Li T-T, Chen X, Huo D, Arifuzzaman M, Qiao S, Jin W-B, Shi H, Li XV, JRI Live Cell Bank Consortium, Iliev ID, *et al* (2024a) Microbiota metabolism of intestinal amino acids impacts host nutrient homeostasis and physiology. *Cell Host Microbe* 32: 661–675.e10
- Liu Y, Liu J, Du M, Yang H, Shi R, Shi Y, Zhang S, Zhao Y & Lan J (2023) Short-chain fatty acid - A critical interfering factor for allergic diseases. *Chem Biol Interact* 385: 110739
- Li X-Y, Zhang S-Y, Hong Y-Z, Chen Z-G, Long Y, Yuan D-H, Zhao J-J, Tang S-S, Wang H & Hong H (2024b) TGR5-mediated lateral hypothalamus-dCA3-dorsolateral septum circuit regulates depressive-like behavior in male mice. *Neuron* 112: 1795–1814.e10
- Loh JS, Mak WQ, Tan LKS, Ng CX, Chan HH, Yeow SH, Foo JB, Ong YS, How CW & Khaw KY (2024) Microbiota-gut-brain axis and its therapeutic applications in neurodegenerative diseases. *Signal Transduct Target Ther* 9: 37
- Mardinoglu A, Shoaie S, Bergentall M, Ghaffari P, Zhang C, Larsson E, Bäckhed F & Nielsen J (2015) The gut microbiota modulates host amino acid and glutathione metabolism in mice. *Mol Syst Biol* 11: 834
- Momozawa Y, Dmitrieva J, Théâtre E, Deffontaine V, Rahmouni S, Charloteaux B, Crins F, Docampo E, Elansary M, Gori A-S, *et al* (2018) IBD risk loci are enriched in multigenic regulatory modules encompassing putative causative genes. *Nat Commun* 9: 2427
- Nishiwaki H, Ito M, Hamaguchi T, Maeda T, Kashihara K, Tsuboi Y, Ueyama J, Yoshida T, Hanada H, Takeuchi I, *et al* (2022) Short chain fatty acids-producing and mucin-degrading intestinal bacteria predict the progression of early Parkinson's disease.

- Roychowdhury T, Klarin D, Levin MG, Spin JM, Rhee YH, Deng A, Headley CA, Tsao NL, Gellatly C, Zuber V, *et al* (2023) Genome-wide association meta-analysis identifies risk loci for abdominal aortic aneurysm and highlights PCSK9 as a therapeutic target. *Nat Genet* 55: 1831–1842
- Wacholder S, Chanock S, Garcia-Closas M, El Ghormli L & Rothman N (2004) Assessing the probability that a positive report is false: an approach for molecular epidemiology studies. *J Natl Cancer Inst* 96: 434–442
- Wang F, Panjwani N, Wang C, Sun L & Strug LJ (2022) A flexible summary statistics-based colocalization method with application to the mucin cystic fibrosis lung disease modifier locus. *American Journal of Human Genetics* 109: 253
- Wellcome Trust Case Control Consortium (2007) Genome-wide association study of 14,000 cases of seven common diseases and 3,000 shared controls. *Nature* 447: 661–678
- Xiong X, Hou L, Park YP, Molinie B, GTEx Consortium, Gregory RI & Kellis M (2021) Genetic drivers of mA methylation in human brain, lung, heart and muscle. *Nat Genet* 53: 1156–1165
- Yang Z, Chen Z, Lin X, Yao S, Xian M, Ning X, Fu W, Jiang M, Li N, Xiao X, *et al* (2022) Rural environment reduces allergic inflammation by modulating the gut microbiota. *Gut Microbes* 14: 2125733
- Zhou M-S, Zhang B, Gao Z-L, Zheng R-P, Marcellin DFHM, Saro A, Pan J, Chu L, Wang T-S & Huang J-F (2021) Altered diversity and composition of gut microbiota in patients with allergic rhinitis. *Microb Pathog* 161: 105272

13th Nov 2025

Manuscript Number: MSB-2025-13191R

Title: Genetics-mediated regulation of intestinal gene expression on microbiome contributes to human disease heritability

Author: Haochuan Wang

Chengyu Li

Zhen Hu

Haonan Feng

Luwei Chen

Ke Ding

Jiuhong Nan

Yuhan Wu

Jinghao Sheng

Xushen Xiong

Dear Prof Xiong,

Thank you for the submission of your revised manuscript to Molecular Systems Biology. We have now received the enclosed report from the reviewers who were asked to re-assess it. As you will see, the reviewers are now all supportive, and I am pleased to inform you that we will be able to accept your manuscript pending the following amendments:

1. Please address the remaining minor comment raised by Reviewer #3.

On a more editorial level, please do the following:

1. Add the missing callouts for Figure 1G.

2. Remove "Author Contribution" section from the manuscript file.

3. Please ensure that the funding information in the manuscript file is consistent with that in the submission system. Currently, the following items are missing from the submission system: Fundamental Research Funds for the Central Universities (no. 226-2025-00176); the Zhejiang Provincial Leading Innovation and Entrepreneurship Team Introduction and Cultivation Program (no. 2024R01024); Liangzhu Laboratory, the State Key Laboratory of Transvascular Implantation Devices, Benyuan Foundation, and K.C.Wong Education Foundation

4. Data availability: code availability should be included into "Data Availability". Remove the heading "Code Availability".

5. "Disclosure statement and competing interests" should be renamed to "DISCLOSURE AND COMPETING INTERESTS STATEMENT".

6. Synopsis image should be sized exactly 550 px wide and 400-600 px high. Ensure that any text on the image remains clear at this size.

7. Please update the source file names, titles, legends, and manuscript callouts to Table EV1-EV5 instead of Dataset EV1, EV5, EV7, EV8, and EV11. The legends should be removed from the manuscript and uploaded above the tables in each corresponding Excel file. The remaining datasets (Dataset EV2-EV4, EV6, EV9-EV10, and EV12-EV14) should be renumbered sequentially as Dataset EV1, EV2, and so on. Their legends should also be removed from the manuscript and uploaded as a separate tab (sheet) within each Excel file.

8. Source Data:

- Source data for main figures should be uploaded as one (zipped) file /figure, and named as "manuscriptID_SourceDataForFigure x"
- All source data for the EV figures should be combined into a single zipped file.

9. Please address the following issues in figure legends:

- Please note that the exact p values are not provided in the legends of figures 3D, EV1 B, EV2B, EV3 C.
- Please indicate the statistical test used for data analysis in the legends of figures 2D, E; 3F, 4A, EV1 B, EV2 F, EV3 E.
- Please note that the box plots need to be defined in terms of minima, maxima, centre, bounds of box and whiskers, and percentile in the legends of figures 5F, EV1 C, D; EV2 B, EV3 A, EV4 G.
- Please note that information related to n is missing in the legends of figures 1D, EV1 C, D; EV2 B, EV3 A.
- Please note that for heatmap present in figures EV4 D, a numbered scale bar is not provided. This needs to be rectified.

- Please note that axis labels are not defined for figure EV2 F.

10. "Materials and methods" should be renamed to "Methods".

11. Sections need to be named and the order should be corrected: Title page - Abstract - Keywords - Introduction - Results - Discussion - Methods - Data Availability - Acknowledgements - Disclosure and Competing Interests Statement - References - Figure Legends - Table(s) - Expanded View Figure Legends.

Click on the link below to submit your revised paper.

Sincerely,
Jingyi

Jingyi Hou, PhD
Senior Editor
Molecular Systems Biology

*** PLEASE NOTE *** As part of the EMBO Press transparent editorial process initiative (see our Editorial at <https://dx.doi.org/10.1038/msb.2010.72> , Molecular Systems Biology will publish online a Review Process File to accompany accepted manuscripts. When preparing your letter of response, please be aware that in the event of acceptance, your cover letter/point-by-point document will be included as part of this File, which will be available to the scientific community. More information about this initiative is available in our Instructions to Authors. If you have any questions about this initiative, please contact the editorial office (msb@embo.org).

Reviewer #1:

The authors have addressed my comments adequately. In my opinion, this is an interesting paper and should be accepted.

Reviewer #2:

I appreciate the authors' detailed and thoughtful revision. They have addressed all my earlier concerns comprehensively, including methodological rigor, sensitivity analyses, and functional validation. The addition of experimental evidence and mechanistic insights further strengthens the manuscript. Overall, I find the revision thorough, well executed, and commend the authors for their careful and substantial improvements.

Reviewer #3:

Haochuan et al. have thoroughly addressed all the concerns I raised in my previous review. The revised manuscript is substantially improved and scientifically sound. I would suggest adding a brief statement in the Discussion regarding population ancestry (as all datasets are of European origin) to clarify the generalizability of the findings. After this minor revision, in my opinion, the manuscript is suitable for publication in Molecular Systems Biology.

Editorial requirements:

1. Please address the remaining minor comment raised by Reviewer #3.

Response: As suggested by Reviewer #3, we have now added a statement in the **Discussion** regarding population ancestry to clarify the generalizability of our results. It reads as:

“The current analyses were restricted to European ancestry due to QTL availability. Cross-ancestry studies may provide a more comprehensive understanding of the ancestry specificity of gene-microbiome crosstalk.”

On a more editorial level, please do the following:

1. Add the missing callouts for Figure 1G.

Response: We have added the callout for Fig. 1G.

2. Remove "Author Contribution" section from the manuscript file.

Response: We have removed this section from our manuscript.

3. Please ensure that the funding information in the manuscript file is consistent with that in the submission system. Currently, the following items are missing from the submission system: Fundamental Research Funds for the Central Universities (no. 226-2025-00176); the Zhejiang Provincial Leading Innovation and Entrepreneurship Team Introduction and Cultivation Program (no. 2024R01024); Liangzhu Laboratory, the State Key Laboratory of Transvascular

Implantation Devices, Benyuan Foundation, and K.C.Wong Education Foundation

Response: We now confirm that the funding information in the manuscript file is consistent with that in the submission system.

4. Data availability: code availability should be included into "Data Availability". Remove the heading "Code Availability".

Response: We have included the code availability into “Data Availability” and removed the heading "Code Availability".

5. "Disclosure statement and competing interests" should be renamed to "DISCLOSURE AND COMPETING INTERESTS STATEMENT".

Response: We have renamed the heading to “DISCLOSURE AND COMPETING INTERESTS STATEMENT”.

6. Synopsis image should be sized exactly 550 px wide and 400-600 px high. Ensure that any text on the image remains clear at this size.

Response: The synopsis image is sized 500 px wide and 400 px high. We confirm that the text on the image is clear at the size.

7. Please update the source file names, titles, legends, and manuscript callouts to Table EV1-EV5 instead of Dataset EV1, EV5, EV7, EV8, and EV11. The legends should be removed from the manuscript and uploaded above the tables in each corresponding Excel file. The remaining datasets (Dataset EV2-EV4, EV6, EV9-EV10, and EV12-EV14) should be renumbered sequentially as Dataset EV1, EV2, and so on. Their legends should also be removed from the manuscript and uploaded as a separate tab (sheet) within each Excel file.

Response: All supplementary files and callouts in the manuscript have been updated accordingly.

8. Source Data:

- Source data for main figures should be uploaded as one (zipped) file /figure, and named as "manuscriptID_SourceDataForFigure x"

- All source data for the EV figures should be combined into a single zipped file.

Response: We have organized the source data for the main figures and EV figures as requested.

9. Please address the following issues in figure legends:

- Please note that the exact *p* values are not provided in the legends of figures 3D, EV1 B, EV2B, EV3 C.

Response: We have added the exact *P* values to the EV1B, EV2B, and EV3C. Of note, the exact *P* values are not available for Fig. 3D, since the *P* values are lower than 2.2×10^{-308} , which is the smallest non-zero floating-point number.

- Please indicate the statistical test used for data analysis in the legends of figures 2D, E; 3F, 4A, EV1 B, EV2 F, EV3 E.

Response: We have added the statistical tests in the Figure legends.

- Please note that the box plots need to be defined in terms of minima, maxima, centre, bounds of box and whiskers, and percentile in the legends of figures 5F, EV1 C, D; EV2 B, EV3 A, EV4 G.

Response: We have defined these statistics for all the boxplots.

- Please note that information related to *n* is missing in the legends of figures 1D, EV1 C, D; EV2 B, EV3 A.

Response: We have added the corresponding number of data points (*n*) to the figures and legends for these figure panels.

- Please note that for heatmap present in figures EV4 D, a numbered scale bar is not provided. This needs to be rectified.

Response: We have added a numbered scale bar for this panel.

- Please note that axis labels are not defined for figure EV2 F.

Response: We have added the y-axis label to this panel.

10. "Materials and methods" should be renamed to "Methods".

Response: We have renamed the section to "Methods".

11. Sections need to be named and the order should be corrected: Title page - Abstract - Keywords - Introduction - Results - Discussion - Methods - Data Availability - Acknowledgements - Disclosure and Competing Interests Statement - References - Figure Legends - Table(s) - Expanded View Figure Legends

Response: We have named and reordered these sections as requested.

We thank the editor for these very specific guidelines for formatting the manuscript files!

Reviewer comments:

Reviewer #1:

The authors have addressed my comments adequately. In my opinion, this is an interesting paper and should be accepted.

Reviewer #2:

I appreciate the authors' detailed and thoughtful revision. They have addressed all my earlier concerns comprehensively, including methodological rigor, sensitivity analyses, and functional validation. The addition of experimental evidence and mechanistic insights further strengthens the manuscript. Overall, I find the revision thorough, well executed, and commend the authors for their careful and substantial improvements.

Reviewer #3:

Haochuan et al. have thoroughly addressed all the concerns I raised in my previous review. The revised manuscript is substantially improved and scientifically sound. I would suggest adding a brief statement in the Discussion regarding population ancestry (as all datasets are of European origin) to clarify the generalizability of the findings. After this minor revision, in my opinion, the manuscript is suitable for publication in Molecular Systems Biology.

Response: As suggested, we have added the statement in the discussion section, which reads as: *"Fourth, the current analyses were restricted to European ancestry due to QTL availability. Cross-ancestry studies may provide a more comprehensive understanding of the ancestry specificity of gene-microbiome crosstalk"*.

We thank these reviewers again for the positive comments, and for the valuable and constructive suggestions throughout the revision, which have helped strengthen this study both technically and mechanistically.

20th Nov 2025

Manuscript number: MSB-2025-13191RR

Title: Genetics-mediated regulation of intestinal gene expression on microbiome contributes to human disease heritability

Dear Prof Xiong,

Thank you again for sending us your revised manuscript. We are now satisfied with the modifications made and I am pleased to inform you that your paper has been accepted for publication.

Sincerely,
Jingyi

Jingyi Hou, PhD
Senior Editor
Molecular Systems Biology
